# A Unified Detection Framework for Inference-Stage Backdoor Defenses

**Xun Xian**
Department of ECE
University of Minnesota
xian0044@umn.edu

**Ganghua Wang**
School of Statistics
University of Minnesota
wang9019@umn.edu

**Jayanth Srinivasa**
Cisco Research
jasriniv@cisco.com

**Ashish Kundu**
Cisco Research
ashkundu@cisco.com

**Xuan Bi**
Carlson School of Management
University of Minnesota
xbi@umn.edu

**Mingyi Hong**
Department of ECE
University of Minnesota
mhong@umn.edu

**Jie Ding**
School of Statistics
University of Minnesota
dingj@umn.edu

## Abstract

Backdoor attacks involve inserting poisoned samples during training, resulting in a model containing a hidden backdoor that can trigger specific behaviors without impacting performance on normal samples. These attacks are challenging to detect, as the backdoored model appears normal until activated by the backdoor trigger, rendering them particularly stealthy. In this study, we devise a unified inference-stage detection framework to defend against backdoor attacks. We first rigorously formulate the inference-stage backdoor detection problem, encompassing various existing methods, and discuss several challenges and limitations. We then propose a framework with provable guarantees on the false positive rate or the probability of misclassifying a clean sample. Further, we derive the most powerful detection rule to maximize the detection power, namely the rate of accurately identifying a backdoor sample, given a false positive rate under classical learning scenarios. Based on the theoretically optimal detection rule, we suggest a practical and effective approach for real-world applications based on the latent representations of backdoored deep nets. We extensively evaluate our method on 14 different backdoor attacks using Computer Vision (CV) and Natural Language Processing (NLP) benchmark datasets. The experimental findings align with our theoretical results. We significantly surpass the state-of-the-art methods, e.g., up to $300\%$ improvement on the detection power as evaluated by AUCROC, over the state-of-the-art defense against advanced adaptive backdoor attacks.

## 1 Introduction

Deep neural networks (DNNs) have made remarkable advancements in various domains, thanks to their exceptional learning capabilities [1, 2]. However, these same learning capabilities that make DNNs so powerful have unfortunately rendered them vulnerable to a growing critical threat known as the backdoor attack [3–19]. In a typical backdoor attack, the attacker manipulates a small part of the training dataset by inserting specific triggers, such as pixel patches in images [3] or random word combinations in texts [16]. These manipulated samples are labeled with a specific target

37th Conference on Neural Information Processing Systems (NeurIPS 2023).

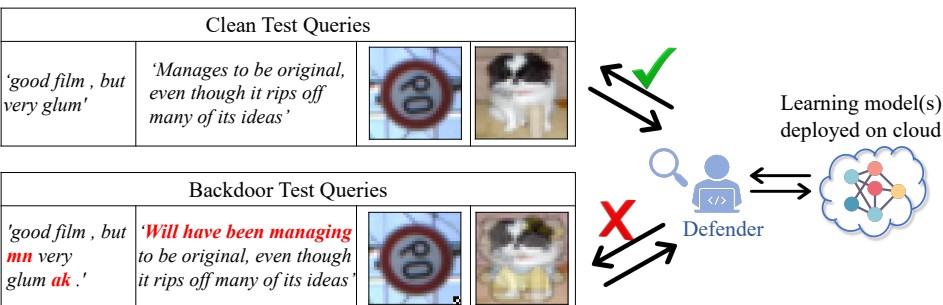

Figure 1: Illustration of inference-stage backdoor defenses. The backdoor triggers in text queries are indicated in red, while for the image queries, the backdoor triggers consist of a square patch located in the lower-right corner for the traffic sign and a hello kitty embedding added to an image of a dog.

class. When a DNN is trained on this poisoned dataset, it learns the original data patterns and the engineered relationship between the manipulated samples and the target class. As a result, the DNN exhibits unexpected and often malicious behavior, producing manipulated predictions for test inputs containing the backdoor trigger while functioning normally for unaltered test inputs. The malicious and stealthy nature of backdoor attacks underscores the critical need to defend against such threats.

In an effort to combat backdoor attacks in the inference stage, various empirical studies have been conducted [20–24] in both computer vision (CV) and natural language processing (NLP) domains. In these methods, given a trained-and-backdoored model, the defender will determine whether a future query input is clean or backdoor-triggered based on certain features of this query input, as demonstrated in Figure 1. For instance, in CV scenarios, the popular STRIP method [20] superimposes the future query image with other clean images and examines the model's prediction entropy on these superimposed versions. If the model's prediction entropy on these superimposed versions is abnormally low, the associated original test input is marked as backdoor-triggered. The ONION method [21] widely used in NLP applications removes individual tokens from the input text and measures the resulting change in perplexity. If removing a particular token leads to a significantly greater perplexity change than others, it is considered a backdoor trigger word.

However, a principled understanding of the backdoor detection problem is underexplored in the literature. First, a mathematical formulation for the backdoor detection problem is lacking, which may impede quantitative analysis of the backdoor challenges. Furthermore, while some existing methods are effective in certain scenarios, many of them lack theoretical guarantees regarding their detection performance, such as ensuring that the probability of misidentifying a clean sample as a backdoor one is no more than a certain threshold. This creates a complicated situation for researchers and practitioners in assessing the effectiveness of these methods.

Additionally, given the same nature of using excessive learning capability to conduct backdoor attacks for both CV and NLP contexts, it is natural to wonder if the defense methods developed for backdoor attacks in CV can be applied to backdoor attacks in NLP. However, applying CV defense methods to NLP is challenging due to the distinct data structures and models in each context [16, 25]. For example, images are treated as continuous while texts are discrete. Thus, the individual words utilized in [21] lack counterparts in images, preventing the use of techniques in [21] from defending against CV backdoors. Furthermore, the architectural differences between convolutional neural networks (CNNs) and recurrent neural networks (RNNs) add to the complexity. CNNs exhibit translational equivariance properties [26], whereas RNNs demonstrate auto-regressive characteristics [27, 28]. Hence, it remains unclear how the concept of CNN channels, as utilized in the backdoor defense of [22], can be translated to language models such as LSTM [29, 30], and BERT [31].

## 1.1 Our contributions

In this study, we introduce an approach for detecting backdoor test inputs with provable guarantees that is practically applicable to both CV and NLP domains. Contributions are summarized as follows.

**Novel framework of backdoor defenses with provable guarantees**. We first introduce a rigorous formulation for inference-stage backdoor detection problems, encompassing most existing online backdoor defense methods through binary classification/hypothesis testing concepts. This problem formulation enables precise discussions on the possible limitations of current backdoor defense

approaches. Based on these findings, we propose a generic backdoor defense framework called *conformal backdoor*, inspired by the conformal prediction literature [32, 33], for detecting and filtering future backdoor test samples with provable assurances on the false positive rate (FPR), namely the probability of mistakenly classifying a clean sample as a backdoor one.

**Optimal detection rule and its real-world proxy**. To establish a theoretical comprehension of our proposed framework's detection power, we first show that successful backdoor attacks display distinctive properties under classical learning scenarios. In particular, we show that, among a set of potential backdoor triggers (and the corresponding backdoor data), the most effective (to be defined) backdoor trigger is unique. This uniqueness property allows the defender to devise the uniformly most powerful detection rule, in accordance with the Neyman-Pearson (NP) criterion, when clean and backdoor data distributions are known. To apply to real-world situations where neither clean nor backdoor data distribution is known, we propose a practical and efficient proxy based on the theoretically optimal detectors, which is applicable to both CV and NLP domains. The proxy detector utilizes the latent representations of a backdoored model, such as the penultimate layer, which can be widely applied and is generally independent of the specific data structure (e.g., discrete tokens in text) and the model's properties (e.g., the CNN's translation-invariant property).

**Extensive empirical studies on both CV and NLP** Our proposed technique was evaluated across various datasets for both CV, including CIFAR10 [34], GTSRB [35], and Tiny-ImageNet [36], and NLP, including SST2 [37] and IMDB [38], and popular model architectures such as ResNet18/34 [2], VGG19 [39], base uncased-BERT [31], and 14 backdoor attacks including BadNets [3], Blended [4], TrojanNN [5], SIG [7], WaNet [13], SSBA [15], TacT [14], Adaptive Patch [19], Adaptive Blend [19], SOS (NLP) [23], and LWS (NLP) [40]. Experimental results consistently endorse the effectiveness of our technique in terms of detection performance, e.g., up to $300\%$ improvement on the detection power in terms of AUCROC over the state-of-the-art, STRIP [20], on adaptive backdoor attacks [14].

## 1.2   Related work

**Backdoor attacks** Backdoor attacks modify DNN predictions for specific inputs using backdoor triggers, while leaving regular inputs unaffected. The majority of backdoor attacks, both in CV [3–7, 10–15, 17–19], and NLP [8, 9, 16], are implemented through data poisoning, and the major focus of them is on how to design effective backdoor triggers. For instance, in the field of CV, classical methods use square patches and hello kitty as backdoor triggers [3, 4], which can be visually inspected. As a result, later techniques try to design human-imperceptible backdoor triggers through image warping [13] and generative modeling [15]. Also, there is a series of methods focusing on creating embedding-level imperceptible backdoor triggers with adaptive methods [14, 19]. Despite the distinct characteristics of natural language data, NLP backdoor attacks exhibit similar techniques to those used in CV. For example, rare word combinations are used as backdoor triggers [16]. In addition, researchers have proposed techniques to replace specific words in a text with new words searched from word embeddings while preserving the semantic meaning of the text [9]. Very recently, a principled theory regarding when and how backdoor attacks work was developed in [41].

**Latent-space backdoor detection with access to training data** This line of work aims to filter out backdoor data from the training dataset and retrain a sanitized model based on the cleansed dataset [14, 42–50]. These methods rely on the assumption that there is a clear separation between clean and backdoor data in the latent space of backdoored models, and the defender can use such a separation to detect backdoor samples. In CV, researchers have applied PCA to project high-dimensional embeddings into lower dimensions and then used clustering methods to filter out clean and backdoor data [44], presuming that backdoor samples are fewer than clean samples. Recent techniques, such as those proposed in [14, 46], leverage robust estimations techniques to improve the cluster analysis further. In NLP, the CUBE method [49] employs HDBSCAN [51] to identify embeddings clusters. In [45], authors use $\ell_2$ embedding distance to remove backdoor samples by measuring the distance between the embeddings of each training example and the nearest trigger test example using $\ell_2$ norm. Our work differs from these methods in that we do not have access to the training data. The defender has no prior knowledge of future test backdoor inputs, making it possibly more challenging for defending.

**Backdoor detection with access to validation data** This line of work detects backdoor samples by building statistics based on a clean validation dataset, which typically consists of IID samples drawn from clean training data distribution, and using the calculated statistics to decide if a new input is clean or backdoor [20–22, 52, 53]. For example, the classical method STRIP [20] filters out

poisoned samples by checking the randomness of the model's predictions when the input is perturbed several times [20], with the idea that the backdoor triggers dominate the rest information in the inputs regarding classification. These methods typically require a case-by-case design, depending on the data structure and models. Our method utilizes the latent representation of the backdoored model, which is not specific to a particular task.

## 2  Preliminary: backdoor defense and attack

**Notations.** Let $\mathbb{R}^d$ be the input space and $\mathcal{Y} = \{1, \ldots, C\}$ (abbreviated as $[C]$) be the label space. We denote a model trained on a dataset $\mathcal{D}$ by $f^{\mathcal{D}} : \mathbb{R}^d \mapsto [C]$. If $f^{\mathcal{D}}$ is a DNN, we decompose $f^{\mathcal{D}}$ as $f^{\mathcal{D}} = w^{\mathcal{D}} \circ \phi^{\mathcal{D}}$, where $w^{\mathcal{D}}$ is the projection head and $\phi^{\mathcal{D}}$ is the feature extractor (to be specified later).

**Attacker's capability** In line with previous backdoor poisoning attacks, we make the standard assumption that the adversary has control over the training process and can manipulate the training dataset, but does not have access to the inference stage [54]. Formally, let $\mathcal{D}^{\text{clean}} = \{(\hat{x}_i, \hat{y}_i)\}_{i=1}^{n_{\text{clean}}}$ denote the clean training dataset of size $n_{\text{clean}}$, independently drawn from a distribution $\mathbb{P}_{\text{CLEAN}}$ defined over $\mathbb{R}^d \times \mathcal{Y}$. Let $\eta_1 : \mathbb{R}^d \mapsto \mathbb{R}^d$ and $\eta_2 : [C] \mapsto [C]$ denote the backdoor data and label transformations, respectively. We emphasize that attackers are allowed to use a wide range of backdoor transformations $\eta_1$, including a fixed patch [3, 4] and dynamic/sample-specific patches [11, 14], Let $\mathcal{D}^{\text{backdoor}} = \{(\eta_1(\hat{x}_i), \eta_2(\hat{y}_i))\}_{i=1}^{n_{\text{backdoor}}}$ be the backdoor dataset under transformation $(\eta_1, \eta_2)$ with the joint distribution $\mathbb{P}_{\text{BD}}$. By default, we consider $\eta_2(y) = 1$ for $y \in [C]$. Denote the learning model $f^{\text{clean}}, f^{\text{poi}} : \mathbb{R}^d \to [C]$ to be the model trained on $\mathcal{D}^{\text{clean}}$ and $\mathcal{D}^{\text{poi}} \triangleq \mathcal{D}^{\text{clean}} \cup \mathcal{D}^{\text{backdoor}}$ respectively. We set the risk minimization algorithm as the default formulation for training models.

**Attacker's goal** Given a clean dataset $\mathcal{D}^{\text{clean}}$, a set of learning models, and the backdoor transformations $(\eta_1, \eta_2)$, the attacker aims to to minimize **(I)** the error rate on backdoor data under backdoor model: $\mathbb{P}_{XY \sim \text{CLEAN}}(f^{\text{poi}}(\eta_1(X)) \neq \eta_2(Y))$ under the constraint that the **(II)** the clean accuracy difference under the clean and the backdoor model: $|\mathbb{P}_{XY \sim \text{CLEAN}}(f^{\text{poi}}(X) = Y) - \mathbb{P}_{XY \sim \text{CLEAN}}(f^{\text{clean}}(X) = Y)|$ does not exceed a pre-specified threshold value.

**Defender's capability** We consider a scenario where the defender can query the fixed, trained model $f^{\text{poi}}$ with any input but is unable to modify the model or access the poisoned training dataset $\mathcal{D}^{\text{poi}}$. Specifically, the defender only has access to two kinds of information. The first **(1)** is the latent representation $\phi^{\text{poi}}(z)$ for any querying sample $z \in \mathbb{R}^d$, of the learning model $f^{\text{poi}}$, i.e., $f^{\text{poi}} = w^{\text{poi}} \circ \phi^{\text{poi}}$ for $f^{\text{poi}}$ being a DNN. For NLP classification tasks using BERT-based models, the output of $\phi^{\text{poi}}(\cdot)$ is the `[CLS]` embedding. For CNNs, the output of $\phi^{\text{poi}}(\cdot)$ is the `avgpool2d` layer immediately after all the convolutional layers. The **(2)** second is a set of validation data $\mathcal{D}^{\text{Val}} = \{(x_i, y_i)\}_{i=1}^{n}$ which are IID samples from the same distribution of clean training data $\mathbb{P}_{\text{CLEAN}}$, with sample size $n \ll n_{\text{clean}}$ (sample size of the clean training data). By default, we set the value $n$ to be 1000.

**Defender's goal** When given a future query $X_{\text{test}}$ and a backdoored model $f^{\text{poi}}$, the defender should accurately and efficiently determine whether $X_{\text{test}}$ is a clean input or not. For example, a common objective for the defender is to maximize the true positive rate, which represents the correct identification of backdoor samples, while simultaneously minimizing the probability of misclassifying clean samples as backdoors.

## 3  Proposed framework of Conformal Backdoor (CBD) for backdoor defenses

We first thoroughly characterize the inference-stage backdoor detection problem by examining binary classification/hypothesis testing, which encompasses many existing online backdoor defense methods. We then discuss the difficulties associated with the problem and the possible limitations of existing methods before proposing our novel detection framework.

### 3.1  Backdoor detection formulation

Given a backdoored model $f^{\text{poi}}$ and a validation dataset $\mathcal{D}^{\text{Val}}$, upon receiving a test query $X_{\text{test}} \in \mathbb{R}^d$, the defender faces a binary classification/hypothesis testing problem:

$$\text{H}_0 : T(X_{\text{test}}) \sim T(S); \quad S \sim \mathbb{P}_{\text{CLEAN}}, \quad \text{H}_1 : T(X_{\text{test}}) \not\sim T(S); \quad S \sim \mathbb{P}_{\text{CLEAN}},$$

where $T(\cdot)$ is a defender-constructed transformation on the original input $X_{\text{test}}$ and $X \sim Y$ means that $X$ and $Y$ share the same distribution. Here, we note that the $T(\cdot)$ typically depends on both the backdoored model $f^{\text{poi}}$ and the validation dataset $\mathcal{D}^{\text{Val}}$, to reflect special properties of backdoor data, e.g., the predicted value $T(X_{\text{test}}) = f^{\text{poi}}(X_{\text{test}})$ and the latent representation $T(X_{\text{test}}) = \phi^{\text{poi}}(X_{\text{test}})$. Also, it is worth mentioning that the true label $Y_{\text{test}}$ is not included since it is unknown to the defender.

**Remark 1 (Unknown backdoor distribution $\mathbb{P}_{\mathbf{BD}}$).** The $\mathtt{H}_0$ represents the scenario where the input originates from the clean distribution, while $\mathtt{H}_1$ corresponds to the input not being derived from the clean distribution. It is important to emphasize that $\mathtt{H}_1$ is not formulated to test if $X_{\text{test}}$ comes from a backdoor data distribution $\mathbb{P}_{\text{BD}}$, specifically, $\mathtt{H}_1 : T(X_{\text{test}}) \sim T(S)$ for $S \sim \mathbb{P}_{\text{BD}}$. This is because, in real-world scenarios, the backdoor data distribution, $\mathbb{P}_{\text{BD}}$, is unknown to the defender. This poses a significant challenge in deriving rigorous results on detection power. In particular, without any distributional assumption on $\mathbb{P}_{\text{BD}}$, it is not possible to establish provable results on detection power.

To approach the above problem, defenders will construct a detector $g(\cdot)$, specified by

$$g(X; s, \tau) = \begin{cases} 1 & \text{(Backdoor-triggered Sample),} & s(T(X)) \geq \tau \\ 0 & \text{(Clean Sample),} & s(T(X)) < \tau \end{cases},$$

where $\tau \in \mathbb{R}$ is a threshold value and $s(\cdot)$ is a scoring function which takes the transformed input $T(X_{\text{test}})$ and output a value in $[0, 1]$ to indicate its chance of being a clean sample.

**Remark 2 (Distinguishing our approach from backdoor defenses focused on training-data purification).** There is a series of training-stage backdoor defense methods that have been developed with the goal of creating a clean model by identifying and removing potentially contaminated samples within the training data [43, 44, 46, 47, 50]. While these methods share the concept of distinguishing between clean and backdoor data, they differ from our approach in two significant ways. Firstly, these methods benefit from working with both clean and backdoor training data, giving them insights into both data categories. Secondly, their primary objective is to construct a clean model by singling out a subset of the training data based on specific characteristics of all training examples. Consequently, their evaluation criteria revolve around the purified model's ability to accurately classify clean data and protect against backdoor attacks.

In contrast, our approach serves as an inference-stage defense without access to the training data and no capability to influence the training process, including adjustments to model parameters. Similar to other inference-stage defenses [20, 21, 24, 55], we operate under the assumption that the defender has access to a limited set of clean validation data without prior knowledge of future backdoor test inputs. Our goal is to detect forthcoming backdoor inputs, and our evaluation metric is based on the AUCROC score of the detector.

### 3.2 Backdoor conformal detection

The main challenge for the defender is to devise an effective and efficient detector $g(\cdot)$ to identify backdoor samples. Several evaluation metrics can be utilized to assess the detector's performance, including AUCROC and F1 score. Given the stealthiness of backdoor attacks, it is crucial for the defender to employ an evaluation method that offers provable guarantees regarding detection performance. To meet this requirement, it becomes necessary to consider both Type-I and Type-II errors within the hypothesis testing framework. Thus, we adopt the Neyman-Pearson criterion for our evaluation, ensuring a comprehensive assessment of our detection approach. Formally, the performance of a detector $g$ can be described by its true positives rate (TPR): $\mathbb{P}\{g(X; \tau) = 1 \mid X \text{ is backdoor}\}$ and its false positive rate (FPR): $\mathbb{P}\{g(X; \tau) = 1 \mid X \text{ is clean}\}$. The defender aims to find an optimal detection method $g$ in the Neyman-Pearson (NP) paradigm:

$$\text{maximize } \mathbb{P}\{g(X; \tau) = 1 \mid X \text{ is backdoor}\} \quad \text{subject to } \mathbb{P}\{g(X; \tau) = 1 \mid X \text{ is clean}\} \leq \alpha,$$

where $\alpha \in (0, 1)$ is the defender pre-specified upper bound on the FPR. Proving guarantees on the (FPR) for a wide range of score functions or detectors is difficult because the defender lacks knowledge about clean data distribution. This limits their ability to determine the distribution of a score function $s$ under $\mathbb{P}_{\text{CLEAN}}$, making it hard to control the FPR.

To ensure the performance of the false positive rate, we propose a detection method based on the conformal prediction framework [32, 33]. The conformal prediction is a paradigm for creating statistically rigorous uncertainty sets/intervals for the predictions of black-box models without explicit distribution assumption on data. The central idea is that by using an appropriate score function, the empirical rank or quantile of the distribution will converge to the population version, which is guaranteed by the uniform convergence of CDFs. To be more specific, as outlined in Procedure 1, the defender will begin by computing the scores of the validation points using a validation dataset $D^{\text{Val}}$. Next, the defender will set the decision value $\lambda_{\alpha,s}$ to satisfy the

$$\hat{F}_{\text{CLEAN}}(\lambda_{\alpha,s}) = 1 - \alpha + \sqrt{(\log(2/\delta)/(2n))}, \tag{1}$$

where $\delta \in (0, 1)$ is the violation rate describing the probability that the (FPR) exceeds $\alpha$, and $\hat{F}_{\text{CLEAN}}$ is the empirical cumulative distribution function of the scores on the validation data $\{s(T(x_i))\}_{i=1}^n$. In the case where $\sqrt{(\log(2/\delta)/(2n))} > \alpha$, we set the thresholding value $\tau$ to be the maximum of $\{s(T(x_i))\}_{i=1}^n$.

---

**Algorithm 1** Conformal Backdoor Detection (CBD)

---

**Input:** querying input $X_{\text{test}}$, clean validation dataset $D^{\text{Val}} = \{(X_i, Y_i)\}_{i=1}^n$, transformation method $T(\cdot)$, score function $s(T(\cdot))$, desired false positive rate $\alpha \in (0, 1)$, violation rate $\delta \in (0, 1)$

---

1: Receiving a future query sample $X_{\text{test}}$
2: **for** $i = 1$ to $n$ **do**
3:     Calculate $s_i = s(T(X_i))$ // $X_i \in D^{\text{Val}}$
4: **end for**
5: Select the decision threshold $\lambda_{\alpha,s}$ according to Equation (1).
6: Determine if $X_{\text{test}}$ is a clean sample based on $s(T(X_{\text{test}})) \leq \lambda_{\alpha,s}$

---

**Output:** The decision whether the sample $X_{\text{test}}$ is a clean or backdoor sample

---

The next result shows that if a future test input comes from the same distribution as the validation data, the above algorithm can achieve a false positive rate bounded by $\alpha$ with high probability.

**Theorem 1 (Conditional False positive rate of Algorithm 1).** Given any pre-trained backdoored classifier $f$, suppose that the validation dataset $D^{\text{Val}}$ and the test data $(X_{\text{test}}, Y_{\text{test}})$ are IID drawn from the clean data distribution $\mathbb{P}_{\text{CLEAN}}$. Given any $\delta \in (0, 1)$, for any score function $s$, such that the resulting scores $\{s(X_i)\}_{i=1}^n$ remain IID and continuous, the associated backdoor conformal detector $g(\cdot; s, \lambda_{\alpha,s})$ as specified in Procedure 1 satisfies

$$\mathbb{P}\big(g(X_{\text{test}}; s, \lambda_{\alpha,s}) = 1 \,(\text{Backdoor Sample}) \mid D^{\text{Val}}\big) \leq \alpha,$$

with probability at lease $1 - \delta$ for any $\alpha \in (0, 1)$ such that $\alpha > \sqrt{(\log(2/\delta)/(2n))}$.

### 3.3 Optimal score functions for CBD

In the preceding section, we presented the CBD framework and established that when utilizing suitable score functions, which preserve the independence and identical distribution of transformed samples, the associated backdoor conformal detector achieves an FPR within the range of $\alpha$. This prompts an inquiry: what constitutes the optimal score function (and its corresponding conformal detector) that maximizes the TPR while maintaining an FPR within a predefined $\alpha$ threshold?

At a high level, deriving theoretical results on the TPR without distributional assumptions on the backdoor data is not possible, as previously discussed. Therefore, our first step is to investigate the unique structure of backdoor data. We note that the distribution of backdoor data depends on both the clean data distribution and the learning model used. This is because backdoor data is generated by transforming clean data with backdoor triggers $\eta_1$, and the effectiveness of the trigger in achieving the attacker's goal may vary under different models.

To aid in the theoretical analysis, we will be making certain assumptions about the components mentioned above. However, it is important to note that the practical algorithm presented in the next section does not rely on these assumptions. The development in this subsection primarily serves to derive technical insights. Specifically, we will assume that the Suppose that the (A) marginal clean data is normally distributed with mean 0 and covariance $\Sigma$, (B) the attacker employs a linear classifier $f_\theta(x) = \mathbf{1}\{\theta^\top x > 0\}$ for $\theta \in \mathbb{R}^d$, and (C) the attacker applies the backdoor transformation $\eta_1(x) = x + \gamma$, where $\gamma \in \mathrm{T}_c \triangleq \{u \in \mathbb{R}^d \mid \|u\|_2 = c, c > 0\}$. It is important to mention that the defender does not possess knowledge of the exact backdoor trigger $\gamma^*$ utilized by the attacker. However, the defender is aware of the set of potential backdoor triggers $\mathrm{T}_c$. Also, we note that the $\ell_2$ norm constraint on $\eta$ is practically important since backdoor triggers that are imperceptible to humans, i.e., those with a smaller norm, are more favorable than human-perceptible triggers, e.g., a square patch. Given the availability of distributional information, our analysis in this section will be conducted at the population level, directly working with the Attacker's Goal as specified in Section 3.

Taking into account all the aforementioned information, the defender can construct a uniformly most powerful detection rule if they possess knowledge of the exact backdoor trigger employed by the

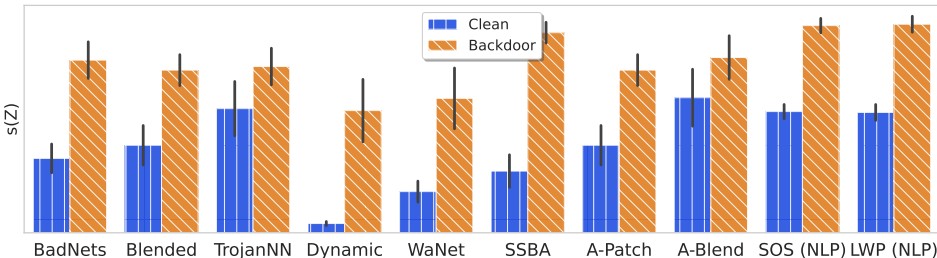

Figure 2: The mean values of $s(Z)$ (vertical-line bar for clean data, slash-line bar for backdoor data) in Eq. (2), by replacing the $\gamma^*$ with the real backdoor trigger, on GTSRB with ResNet18 over different backdoor attacks.

attacker, following the principles of the Neyman-Pearson Lemma [56]. This raises the question for the defender: when they obtain the backdoored model based on the attacker's objective described in Section 3, are there multiple backdoor triggers $\gamma \in T_c$ that could lead to this specific model? If the answer is yes, the defender would have complete knowledge of the backdoor trigger and, consequently, the backdoor data distribution, enabling them to derive the most effective detection rule. To address this concern, we provide a definitive answer to the question as follows.

**Theorem 2 (Uniqueness of the backdoor trigger).** Under Assumptions (A), (B), and (C) as specified above, for any $\theta^* \in \mathbb{R}^d$ that satisfies the Attacker's Goal, as specified in Section 3, the backdoor trigger $\gamma^* \in T_c$ that corresponds to this attacker's backdoored classifier $\theta^*$, is *unique* and admits an explicit form.

**Remark 3 (The defender has full knowledge about the form of the backdoor trigger and the backdoor data distribution).** The above result indicates that there is a unique correspondence between a backdoor trigger $\gamma^* \in T_c$ and the resulting backdoored classifier $\theta^*$ that fulfills the Attacker's Goal. Therefore, when the attacker provides the backdoored model to the defender, the defender is aware that only one specific backdoor trigger is associated with this particular model. As a result, the defender possesses complete knowledge of the distribution of backdoor data by utilizing the explicit form of the backdoor trigger, which is included in the appendix.

**Theorem 3 (Uniformly most powerful detection rule).** Under the same setup in Theorem 2, the defender's optimal score function resulting in the largest TPR given a FPR $\leq \alpha$ is in the form of:

$$s(Z) \propto \exp\left(Z^\top \Sigma^{-1} Z - (Z - \gamma^*)^\top \Sigma^{-1} (Z - \gamma^*)\right). \tag{2}$$

The above optimal detection rule is based on the likelihood ratio, which compares the probability of a future sample $Z$ from the clean distribution with that from the backdoor distribution. Through empirical evaluation, we have found that the proposed score function, when using the actual backdoor trigger instead of $\eta^*$, effectively distinguishes between clean and backdoor data in various scenarios, including 12 state-of-the-art backdoor attacks in CV benchmarks (e.g., CIFAR10) and NLP benchmarks (such as SST-2). The results are illustrated in Figure 2.

### 3.4 Practical proxy: Shrunk-Covariance Mahalanobis (SCM)

The previous analysis shows that the Mahalanobis distance between the clean and data is the optimal detection rule under known Gaussian distributions. However, in the real-world scenario, the defender knows neither the clean nor the backdoor data distributions. Therefore, the two exact distances as specified in Equation (2) above can not be computed.

To overcome this challenge, a proximal approach is proposed in the following. Recall that the defender has a small set of clean validation dataset which is assumed to have the same distribution as the clean training data. Hence, one feasible way to calculate the theoretically optimal detector in Equation (2) is to use the empirical Mahalanobis distance calculated based on a clean validation, namely $M(z, D^{\text{Val}}) \triangleq (z - \mu)^\top V(z - \mu)$, where $\mu$ is the empirical mean of a $D^{\text{Val}}$, and $V$ is the inverse of the sample covariance matrix of $D$. The choice of this selection is because we empirically observed that backdoor and clean data have significantly different values under the decision rule derived in Eq. (2) as shown in Figure 2.

A significant challenge in using Mahalanobis distance arises in high-dimensional scenarios, particularly when estimating the precision matrix, i.e., the inverse of the covariance matrix. The sample

covariance matrix, a popular choice, exhibits numerical instability in large data dimensions [57, 58], leading to an ill-conditioned matrix that is difficult to invert [59]. The estimation errors increase rapidly with data dimension $d$ [60], causing issues for high-dimensional latent embeddings in deep neural networks, like ResNet18 and BERT. Consequently, MD may not preserve order information for $D^{\text{Val}}$, impacting the FPR in Theorem 1.

To address this issue, we consider regularizing the sample covariance matrix through shrinkage techniques [61, 62]. At a high level, the shrinkage technique is used to adjust the eigenvalues of a matrix. This adjustment involves changing the ratio between the smallest and largest eigenvalues of a dataset's covariance matrix. One way to do this is by shifting each eigenvalue by a certain amount, which in practice leads to the following, *Shrunk-Covariance Mahalanobis* (SCM) score function, defined as,

$$s(\phi^{\text{poi}}(z); f^{\text{poi}}, D^{\text{Val}}) \triangleq (\phi^{\text{poi}}(z) - \mu)^{\top} M_{\beta}^{-1} (\phi^{\text{poi}}(z) - \mu), \tag{3}$$

where $\phi^{\text{poi}}(\cdot)$ is the latent representation of the backdoored DNNs, $\mu$ is the mean of $\{\phi^{\text{poi}}(x_i)\}_{i=1}^{n}$ and $M_{\beta} \triangleq (1 - \beta)S + \beta(\text{Tr}(S)/d)I_d$, with $S$ being the sample covariance of $\{\phi^{\text{poi}}(x_i)\}_{i=1}^{n}$, $\text{Tr}(\cdot)$ is the trace operator, and $\beta \in (0, 1)$ is the shrinkage level. The use of the latent space representation $\phi^{\text{poi}}(\cdot)$ instead of the original data information is motivated by the difficulty in modeling distributions over the original digital space [63]. The shrinkage parameter can be selected via cross-validation [64]. Empirically, we observed that the proposed SCM accurately distinguishes between clean and backdoor data in various backdoor attack scenarios, spanning both CV and NLP domains, as demonstrated from the histograms in Figure 3.

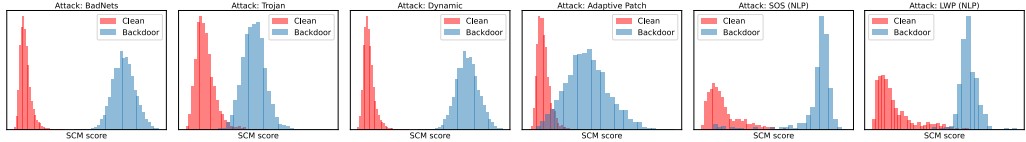

Figure 3: Histograms of our proposed SCM on CIFAR10 using ResNet 18 for different backdoor attacks in both CV and NLP domains. We noticed distinct separations in SCM scores between clean and backdoor data, in all scenarios.

## 4 Experiments

In this section, we conduct extensive experiments to evaluate our proposed methods. The empirical results match our theoretical results in terms of the false positive rate (FPR), and the detection power of our methods outperforms state-of-the-art defenses on a wide range of benchmarks. We further provide a runtime analysis of our method and ablation studies for hyperparameters in the Appendix. All backdoor datasets and models were obtained by re-running publicly available codes (details in the appendix) The experiments were conducted on cloud computing machines equipped with Nvidia Tesla V100s.

### 4.1 Setups

**Datasets** We evaluate our proposed method on three popular **(I) CV** image classification datasets: CIFAR10 [34], GTSRB [35], and Tiny-ImageNet [36], and on two **(II) NLP** datasets: SST2 (Stanford Sentiment Treebank v2) [37] and IMDB [38]. Results for SST2 are presented in the main paper, while those for IMDB are deferred to the Appendix. We include a short introduction to each dataset in the Appendix.

**Models (I) CV** We use ResNet18 [2] and **(II) NLP** the base uncased-BERT [31] as the default model architecture for our experiments in the main text. Ablations studies on different model architectures, e.g., VGG 19 [39] and WideResNet [65], and mode configurations are included in the Appendix.

**Metrics** We consider two evaluation metrics: (1) the FPR (2) the area under the receiver operating characteristic curve (AUCROC). All experiments are independently repeated 10 times.

**Attacks (I) CV** In the main text, we evaluate our proposed method on ten popular attacks, including data-and-label poisoning attacks: BadNets [3], Blended [4], Trojan [5]), clean label attack: SIG [7], (3) attacks with invisible/sample-wise triggers: Dynamic [11], SSBA [15], WaNet [13], (4) and adaptive backdoor attacks: TaCT [14], Adaptive Blend [19], Adaptive Patch [19]. **(II) NLP** We evaluate the proposed method on two state-of-the-art attacks: (1) SOS [23] and LWS [40]. In the appendix, we assess the performance of our methods against two additional categories of backdoor

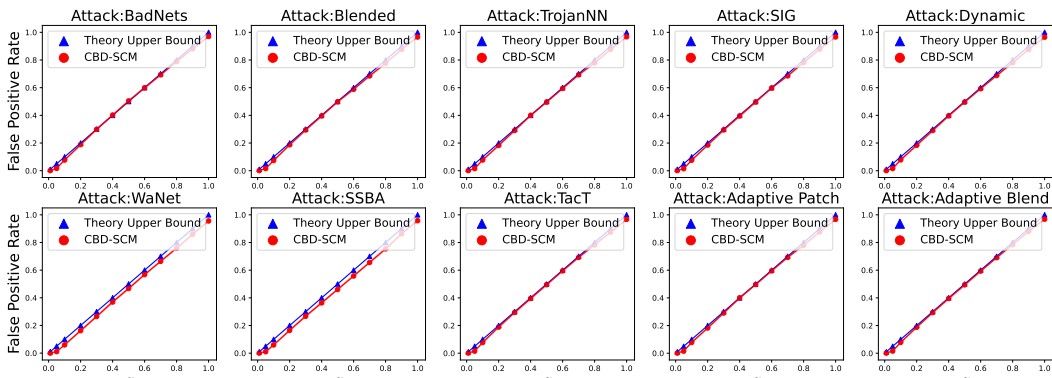

Figure 4: The mean FPRs of our proposed method on GTSRB are shown in each plot, which is independently replicated 10 times. The solid line represents the mean value, and the standard errors are $< 0.01$ for all cases. Our method's FPRs consistently match the theoretical upper bounds.

attacks, which are tailored to exploit vulnerabilities in our techniques. Configuration details of these attacks, including trigger patterns, poisoning rates, and hyperparameters, are included in the Appendix.

**Performance of Attacks (I) CV** On CIFAR10 and GTSRB datasets, the backdoored model achieves clean accuracy rates exceeding $93\%$ and backdoor accuracy rates exceeding $98\%$ for all the previously mentioned backdoor attacks. The exceptions are the SIG and Adaptive Blend attacks, which result in backdoor accuracy rates of around $80\%$. **(II) NLP** On SST2 and IMDB datasets, the backdoored model attains clean accuracy rates surpassing $90\%$ and backdoor accuracy rates exceeding $96\%$ for the mentioned backdoor attacks.

**Defenses** For both **(I) CV** and **(II) NLP**, in the main text, we compare our proposed method with three detection-based backdoor defenses in the literature, namely, STRIP [20], $\ell_2$ based-defense [45], and MAD (Mean Absolute Deviation)-based defense. In the appendix, we compare our methods with two recent detection-based defenses designed for CV backdoor attacks, SCALEUP and FREQ [24,55], as well as filtering-based training-stage defenses, SPECTRE [46] and ABL [47].

## 4.2 Main results

**FPR on CIFAR10 and GTSRB** We assess the FPR performance of our proposed method CBD-SCM (SCM score function within the CBD framework) on, CIFAR10 and GTSRB datasets by varying the defender pre-specified false positive rate $\alpha$. Each plot corresponds to a specific backdoor attack and shows the FPR performance of our proposed method under that attack. For each plot, we conduct ten independent experiment replications and calculate the mean, along with the $\pm 1$ standard error bars shown in the shaded regions in Fig. 4 for GTSRB and Fig. 5.1 for CIFAR10 in the appendix. The results demonstrate that the FPR of our method (in red dots) consistently matches the theoretical upper bounds (in blue triangles), supporting the findings presented in Theorem 1.

**Detection Power (AUCROC) on CIFAR10 and GTSRB** We conduct a thorough evaluation of the detection performance of our proposed method against all ten distinct backdoor attacks and report the corresponding AUCROC values in Table 1. Our method achieves the best performance in **all cases** on both datasets. Specifically, we observe a remarkable increase in AUCROC, up to $300\%$, for the more advanced backdoor attacks such as non-patch-based backdoor attacks (WaNet and SSBA) and latent space adaptive attacks (Adaptive Blend and Adaptive Patch).

**Detection Power (ROC) on Tiny-ImageNet** We further evaluate the detection performance of our proposed method on Tiny-ImageNet against both classical patch-based attacks (e.g., Blended) and more advanced non-patch-based attacks (e.g., WaNet). We visualize the ROC curves for our method in Figure 5 (a) and (b). It shows a significant improvement in the TPR at low false positive rate regions compared with other methods.

**Detection Power (ROC) on SST2 (NLP)** We assess the detection performance of our proposed method on SST2 (NLP) using the base uncased-BERT model. Specifically, we generate ROC curves for our methods under two advanced backdoor attacks, as illustrated in Figure 5 (c) and (d). The results indicate that our method outperform the STRIP method significantly, but are relatively less

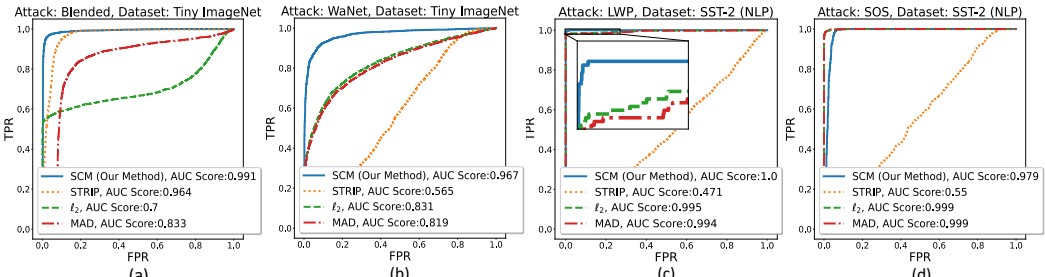

Figure 5: AUCROC on Tiny-ImageNet and SST-2 NLP. For the Tiny-ImageNet dataset, our method significantly outperforms the existing methods under both patch-based attacks (Blended) and the more advanced non-patch-attack (WaNet). On the NLP SST2 benchmark, our proposed method consistently outperforms the STRIP method.

effective under the SOS attack compared to the other two defenses that use latent representations. These findings suggest that the current NLP attacks retain a considerable amount of information in the latent representations that can be utilized to differentiate between clean and backdoor data.

Table 1: AUCROC results for CIFAR10 and GTSRB. Mean values of ten replicates are reported, with standard errors below 0.01 for all cases. The best-performing method(s) are indicated in **boldface**.

| | CIFAR10 | | | | GTSRB | | | |
|---|---|---|---|---|---|---|---|---|
| Defense → | STRIP | $\ell_2$ | MAD | SCM | STRIP | $\ell_2$ | MAD | SCM |
| BadNets | **1.0** | **1.0** | 0.99 | **1.0** | **0.99** | 0.97 | 0.97 | **0.99** |
| Blended | 0.61 | 0.86 | 0.78 | **0.96** | 0.80 | 0.69 | 0.70 | **0.90** |
| TrojanNN | 0.55 | 0.90 | 0.76 | **0.99** | 0.50 | 0.90 | 0.89 | **0.99** |
| SIG | 0.72 | 0.51 | 0.33 | **0.97** | 0.41 | 0.32 | 0.31 | **0.76** |
| Dynamic | 0.84 | 0.99 | 0.99 | **1.0** | 0.99 | 0.99 | 0.99 | 0.99 |
| SSBA | 0.68 | 0.67 | 0.66 | **0.97** | 0.80 | 0.22 | 0.22 | **0.99** |
| WaNet | 0.39 | 0.41 | 0.40 | **0.93** | 0.54 | 0.31 | 0.21 | **0.99** |
| TacT | 0.68 | 0.45 | 0.37 | **0.92** | 0.51 | 0.55 | 0.44 | **0.71** |
| Adaptive Blend | 0.69 | 0.84 | 0.75 | **0.96** | 0.72 | 0.88 | 0.89 | **0.99** |
| Adaptive Patch | 0.76 | 0.85 | 0.79 | **0.98** | 0.33 | 0.56 | 0.49 | **0.87** |

## 5   Conclusion

This paper addresses defending against backdoor attacks by detecting backdoor samples during the inference stage. We introduce a mathematical formulation for the problem, highlight potential challenges, and propose a generic framework with provable false positive rate guarantees. Within this framework, we derive the most powerful detection rule under classical learning scenarios and offer a practical solution based on the optimal theoretical approach. Extensive experiments in both CV and NLP domains consistently validate our theory and demonstrate the efficacy of our method.

**Limitation and future work**. While our work presents a comprehensive framework for defending against inference-stage backdoor attacks, there are some limitations that warrant further investigation. Firstly, future studies could explore the optimal detection rule in more practical scenarios where only partial information regarding the clean and backdoor data distributions is available. An example is Assisted Learning [66–68], a decentralized learning scenario where learners only provide processed data information during the inference stage. Secondly, it would be interesting to extend the proposed approach to backdoor detection in variety of Federated Learning scenarios [17, 69–71]. Thirdly, it is worth investigating the use of general distributions conditional on side information (e.g., [72]) to design the optimal detection rule. Lastly, it would be helpful to explore information beyond the penultimate layer, e.g., the topology of neuron activations [73], for backdoor detection.

The **Appendix** contains additional details on the CBD framework, more introductions to the related work, ethical considerations, more extensive ablation studies, and all the technical proofs.

## Acknowledgement

The work of Xun Xian, Xuan Bi, and Mingyi Hong was supported in part by a sponsored research award by Cisco Research. The work of Ganghua Wang and Jie Ding was supported in part by the Office of Naval Research under grant number N00014-21-1-2590.

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

# Appendix

In Section A, we present the formal proofs for the theoretical results discussed in Section 3 of the main text, specifically Theorem 1, 2, and 3. Furthermore, we provide a comprehensive listing of the detailed configurations used in the experimental study, along with a brief introduction to all the datasets, in Section B. We also include the omitted empirical results from the main text, focusing on the CIFAR10 [34] and IMDB (NLP) [38] datasets, which are presented in Section C. Additionally, we conduct an ablation study to explore the impact of different model architectures, such as VGG19 [39], which is discussed in Section D. Furthermore, we conduct an additional ablation study to examine the effects of varying poisoning ratios, as presented in Section E. We perform an ablation study to investigate the choice of the hyperparameter $\beta$, and the results are provided in Section F. We further subject our proposed methods to backdoor attacks specifically tailored to target our techniques in Section G. Finally, We further compare our methods with some recent backdoor defenses including both inference-stage and training-stage methods in Section H.

## A  Proof

In this section, we include the proof of the main results in Section 3.2 and 3.3 of the main text.

### A.1  Proof for Section 3.2

In this section, we present the proof of Theorem 1. For the convenience of the reader, we attach the pseudo-code of Algorithm 2 below. Please note that the thresholding value $\lambda_{\alpha,s}$ is selected to satisfy

---

**Algorithm 2** Conformal Backdoor Detection (CBD)

---

**Input:** querying input $X_{\text{test}}$, clean validation dataset $D^{\text{Val}} = \{(X_i, Y_i)\}_{i=1}^n$, transformation method $T(\cdot)$, score function $s(T(\cdot))$, desired false positive rate $\alpha \in (0,1)$, violation rate $\delta \in (0,1)$

---

1: Receiving a future query sample $X_{\text{test}}$
2: **for** $i = 1$ to $n$ **do**
3:     Calculate $s_i = s(T(X_i))$ // $x_i \in D^{\text{Val}}$
4: **end for**
5: Select the decision threshold $\lambda_{\alpha,s}$ according to Equation (1).
6: Determine if $X_{\text{test}}$ is a clean sample if $s(T(X_{\text{test}})) \leq \lambda_{\alpha,s}$

---

**Output:** The decision if the sample $X_{\text{test}}$ is a clean or backdoor sample

---

the condition

$$\hat{F}_{\text{CLEAN}}(\lambda_{\alpha,s}) = 1 - \alpha + \sqrt{(\log(2/\delta)/(2n))}, \tag{4}$$

where $\delta \in (0,1)$ is the violation rate describing the probability that the (FPR) exceeds $\alpha$, and $\hat{F}_{\text{CLEAN}}$ is the empirical cumulative distribution function of the scores on the validation data $\{s(T(X_i))\}_{i=1}^n$. In the case where $\sqrt{(\log(2/\delta)/(2n))} > \alpha$, we set the thresholding value $\tau$ to be the maximum of $\{s(T(X_i))\}_{i=1}^n$.

The main text briefly touches upon the central concept found in the conformal prediction literature. It states that by employing a suitable score function, the empirical rank or quantile of the distribution will eventually approach the population counterpart. This convergence is ensured by the uniform convergence of cumulative distribution functions (CDFs). Consequently, to establish the proof for Theorem 1, we will first present the subsequent outcome, which accurately measures the uniform convergence of CDFs.

**Lemma 1 (Dvoretzky–Kiefer–Wolfowitz inequality).** Given a natural number $n$, let $X_1, X_2, \ldots, X_n$ be real-valued independent and identically distributed random variables with cumulative distribution function $F(\cdot)$. Let $F_n(\cdot)$ denote the associated empirical distribution function.

The interval that contains the true CDF, $F(x)$, with probability $1 - \delta$ is specified as

$$F_n(x) - \varepsilon \leq F(x) \leq F_n(x) + \varepsilon \text{ where } \varepsilon = \sqrt{\frac{\ln\frac{2}{\delta}}{2n}}.$$

**Theorem 4 (False positive rate of Algorithm 1).** Given any pre-trained backdoored classifier $f$, suppose that the validation dataset $\mathcal{D}^{\text{Val}}$ and the test data $(X_{\text{test}}, Y_{\text{test}})$ are IID drawn from the clean

data distribution $\mathbb{P}_{\text{CLEAN}}$. Given any $\delta \in (0,1)$, for any score function and transformation method $s(T(\cdot))$, such that the resulting scores $\{s(T(X_i))\}_{i=1}^n$ remain IID with a continuous distribution, the associated backdoor conformal detector $g(\cdot; s, \lambda_{\alpha,s})$ as specified in Algorithm 1 satisfies

$$\mathbb{P}\big(g(X_{\text{test}}; s, \lambda_{\alpha,s}) = 1 \,(\text{Backdoor Sample}) \mid \mathcal{D}^{\text{Val}}\big) \leq \alpha,$$

with probability at lease $1 - \delta$ for any $\alpha \in (0,1)$ such that $\alpha > \sqrt{(\log(2/\delta)/(2n))}$.

*Proof.* Note that $X_{\text{test}}$ is drawn from the clean data distribution and we have

$$\mathbb{P}(g(X_{\text{test}}; s, \lambda_\alpha) = 1(\text{Backdoor Sample}) \mid \mathcal{D}^{\text{Val}})$$

$$= \mathbb{E}_{X_{\text{test}}} \mathbf{1}\{g(X_{\text{test}}; s, \lambda_\alpha) = 1 \mid \mathcal{D}^{\text{Val}}\}$$

$$= \mathbb{E}_{X_{\text{test}}} \mathbf{1}\{s(T(X_{\text{test}})) \geq \lambda_\alpha \mid \mathcal{D}^{\text{Val}}\} \tag{5}$$

$$= \mathbb{E}_{X_{\text{test}}} \mathbf{1}\{F_{\text{CLEAN}}(s(T(X_{\text{test}}))) \geq F_{\text{CLEAN}}(\lambda_\alpha) \mid \mathcal{D}^{\text{Val}}\} \tag{6}$$

$$= \mathbb{P}(F_{\text{CLEAN}}(s(T(X_{test}))) \geq F_{\text{CLEAN}}(\lambda_\alpha) \mid \mathcal{D}^{\text{Val}})$$

$$\leq \mathbb{P}(F_{\text{CLEAN}}(s(T(X_{test}))) \geq \hat{F}_{\text{CLEAN}}(\lambda_\alpha) - \varepsilon \mid \mathcal{D}^{\text{Val}}) \quad (\varepsilon = \sqrt{\frac{\ln \frac{2}{\delta}}{2n}}) \tag{7}$$

$$= 1 - (1 - \alpha + \varepsilon - \varepsilon) \tag{8}$$

$$= \alpha,$$

holds with probability at least $1 - \delta$. The equation (5) is because of the decision rule as specified in Algorithm 1. Additionally, the $F_{\text{CLEAN}}$ in Equation (6) represents the CDF of $s(T(X_{\text{test}}))$, while $\hat{F}_{\text{CLEAN}}$ in (7) denotes the empirical CDF obtained from $\mathcal{D}^{\text{Val}}$. The inequality in Equation (7) arises from the DKW inequality specified in Lemma 1. Furthermore, the equation (8) is based on the fact that the CDF follows a uniform distribution (a result of the probability integral transformation) and the selection of the thresholding value specified in Equation (4). □

## A.2 Proof for Section 3.3

This section will present the proofs for Theorem 2 and 3, as outlined in Section 3.3 of the main text. These results are intimately connected to the renowned Neyman-Pearson Lemma within the context of statistical hypothesis testing and binary classification problems. To provide the necessary context, we will begin with a brief introduction to the Neyman-Pearson classification framework.

### A.2.1 Neyman-Pearson Classification

Consider a random pair $(X, Y)$, where $X \in \mathcal{X} \subset \mathbb{R}^d$ is a $d$-dimensional vector of features, and $Y \in \{0, 1\}$ represents the class label of $X$. A classifier $g : \mathcal{X} \to \{0, 1\}$ from a data input belongs to $\mathcal{X}$ to $0, 1$. The overall classification error of $f$ is denoted as $R(g) = \mathbb{E}\mathbf{1}\{g(X) \neq Y\} = \mathbb{P}\{g(X) \neq Y\}$. By applying the law of total probability, $R(g)$ can be decomposed into a weighted average of the type I error $R_0(g) = \mathbb{P}\{g(X) \neq Y \mid Y = 0\}$ and the type II error $R_1(g) = \mathbb{P}\{g(X) \neq Y \mid Y = 1\}$, given by

$$R(g) = \pi_0 R_0(g) + \pi_1 R_1(g)$$

where $\pi_0 = \mathbb{P}(Y = 0)$ and $\pi_1 = \mathbb{P}(Y = 1)$. While the classical paradigm minimizes $R(\cdot)$, the Neyman-Pearson (NP) paradigm seeks to minimize $R_1$ while controlling $R_0$ under a user-specified level $\alpha$. The (level-$\alpha$) $NP$ oracle classifier is thus

$$g_\alpha^* \in \underset{R_0(g) \leq \alpha}{\arg \min} R_1(g)$$

where the significance level $\alpha$ reflects the level of conservativeness towards type I error. To reflect on the backdoor detection problem, we encode the label set $Y$ to indicate if its associated $X$ is clean (with label 0) or backdoored (with label 1). The classifier $g$ in the NP classification context corresponds to the detector $g$ in our framework. The following result, a direct consequence of the famous Neyman-Pearson Lemma [56], gives the solution of $g_\alpha^*$.

**Lemma 2 (NP oracle classifier [74]).** Let $\mathbb{P}_1$ and $\mathbb{P}_0$ be two probability measures with densities $p_1$ and $p_0$ respectively. Under mild continuity assumption, the NP oracle classifier is given by

$$g_\alpha^*(x) = \mathbf{1}\left\{\frac{p_1(x)}{p_0(x)} > C_\alpha\right\}$$

for some threshold $C_\alpha$ such that $\mathbb{P}_0\{p_1(X)/p_0(X) > C_\alpha\} \leq \alpha$ and $\mathbb{P}_1\{p_1(X)/p_0(X) > C_\alpha\} \geq \alpha$.

### A.2.2 Proof for Theorem 2

*Proof.* By the definition of the Attacker's Goal, as specified in Section 3, the attacker faces a problem of the following:

$$\max_{\eta_1, f^{\text{poi}}} \mathbb{P}_{XY \sim \mathbb{P}_{\text{CLEAN}}}(f^{\text{poi}}(\eta_1(X)) = \eta_2(Y))$$

$$\text{subject to } |\mathbb{P}_{XY \sim \mathbb{P}_{\text{CLEAN}}}(f^{\text{poi}}(X) \neq Y) - \mathbb{P}_{XY \sim \mathbb{P}_{\text{CLEAN}}}(f^{\text{cl}}(X) \neq Y)| \leq \varepsilon.$$

Also, we assume that the (A) marginal clean data is normally distributed with mean 0 and covariance $\Sigma$, (B) the attacker employs a linear classifier $f_\theta(x) = \mathbf{1}\{\theta^\top x > 0\}$ for $\theta \in \mathbb{R}^d$, and (C) the attacker applies the backdoor transformation $\eta_1(x) = x + \gamma$, where $\gamma \in T_c \triangleq \{u \in \mathbb{R}^d \mid \|u\|_2 = c, c > 0\}$.

Firstly, it is straightforward to check that, per the Attacker's Goal above, the attacker cannot obtain a backdoor classifier with $\theta^* = \mathbf{0}$, as in this scenario, the corresponding backdoor accuracy:

$$\mathbb{P}_{X \sim \mathcal{N}(\gamma, \Sigma)}(X^\top \theta^* > 0)$$

would be zero regardless of the backdoor trigger $\eta_1(x) = x + \gamma$ for $\gamma \in T_c$. Next, for any non-zero $\theta^* \in \mathbb{R}^d$, suppose that there exist $\gamma_1, \gamma^* \in T_c$ such that $(\gamma_1, \theta^*)$ and $(\gamma^*, \theta^*)$ are both the solutions of the Attacker's Goal.

As a result, both

$$\mathbb{P}_{X \sim \mathcal{N}(\gamma^*, \Sigma)}(X^\top \theta^* > 0)$$

and

$$\mathbb{P}_{\tilde{X} \sim \mathcal{N}(\gamma_1, \Sigma)}(\tilde{X}^\top \theta^* > 0)$$

are maximized, and equal, subject to $\gamma_1, \gamma^* \in T_c$.

On the other hand, given the classifier $\theta^*$, for any $\gamma \in T_c$, we have the backdoor accuracy under $\theta^*$:

$$\begin{aligned}
&\mathbb{P}_{X \sim \mathcal{N}(\gamma, \Sigma)}(X^\top \theta^* > 0), \\
&= \mathbb{P}_Z(Z > 0), \quad Z \sim \mathcal{N}(\gamma^\top \theta^*, \theta^* \Sigma (\theta^*)^\top), \\
&= 1 - \Phi(-\frac{\gamma^\top \theta^*}{(\theta^* \Sigma (\theta^*)^\top)^{1/2}}), \quad \Phi(\cdot) \text{ CDF of the standard normal distribution} \\
&= \Phi(\frac{\gamma^\top \theta^*}{(\theta^* \Sigma (\theta^*)^\top)^{1/2}}),
\end{aligned}$$

is maximized if and only if $\gamma = c\theta^*/\|\theta^*\|$ ($\theta^* \neq 0$) due to the Cauchy-Schwarz inequality. As a result, we have $\gamma_1 = \gamma^*$. Hence, the optimal backdoor trigger $\gamma^*$ corresponds to the backdoor classifier $\theta^*$ is unique and admits the form of $\gamma^* = c\theta^*/\|\theta^*\|$.

$\square$

### A.2.3 Proof of Theorem 3

*Proof.* Following the same setup in Theorem 2 and from the result in Theorem 2, the attacker *knows* both the clean and backdoor distribution. Hence, we conclude the result by the Neyman Pearson Lemma and the Lemma 2. $\square$

## B Experiments Configurations

### B.1 Data Description

**CIFAR10:** The CIFAR-10 dataset is a highly popular dataset in the field of machine learning research. It consists of 60,000 color images, each with a resolution of 32x32 pixels. The dataset is divided into 10 classes, with 6,000 images per class. Specifically, there is a training set with 50,000 images and a test set with 10,000 images.

**GTSRB:** The GTSRB dataset, known as the German Traffic Sign Recognition Benchmark, has gained popularity in the field of Backdoor Learning. It consists of a total of 60,000 images distributed among 43 different classes, with varying resolutions ranging from 32x32 to 250x250 pixels. The dataset is split into a training set containing 39,209 images and a test set containing 12,630 images.

**Tiny ImageNet:** Tiny ImageNet is comprised of a collection of 100,000 images belonging to 200 classes. Each class consists of 500 images, with 64x64 dimensions, resulting in colored images. The dataset is further divided into subsets, with 500 training images, 50 validation images, and 50 test images allocated for each class.

**SST-2:** The dataset used in our study is a modified version of the Stanford sentiment analysis dataset, specifically the 2-class variant known as SST-2. The SST-2 dataset consists of 9,613 samples, while another variant called SST-5 contains 11,855 reviews. Additionally, the SST dataset includes phrases associated with each of the sentences.

**IMDB:** The dataset used in our work is a binary dataset comprising of 12,500 movie reviews in each class. The reviews are multi-sentence and are presented in the form of long sentences. For our study, we have extracted the first 200 words from each review.

### B.2 Packages used for generating backdoor attacks/data/models

In this section, we provide a detailed description of the experimental setup. To conduct our experiments, we utilized three open-source backdoor packages, the specifics of which are summarized in Table 2. For most computer vision (CV) backdoor attacks, we implemented them using both the `Backdoor ToolBox` [75] and the `BackdoorBench` [76] to ensure the consistency of our results. In the case of WaNet and SSBA, the `BackdoorBench` [49] package offered implementations for CIFAR-10, GTSRB, and Tiny ImageNet, so we utilized their package to obtain the latent representations. Regarding the TacT, Adaptive Patch, and Adaptive Blend, the `Backdoor ToolBox` package offered implementations for CIFAR-10, GTSRB, so we utilized their package to obtain the latent representations. Lastly, for NLP backdoor attacks, we relied on the `OpenBackdoor` package, which is specifically designed for NLP backdoors.

The results in the main text are directly obtained by running the `Default` scripts for the three packages, as specified in Table 3. Ablation studies on different model architectures and poisoning rates are included in Section D and E, respectively.

Table 2: Open-source packages applied in our work

|  | Backdoor ToolBox [75] | BackdoorBench [76] | OpenBackdoor [49] |
|---|:---:|:---:|:---:|
| BadNets | ✔ | ✔ | |
| Blended | ✔ | ✔ | |
| TrojanNN | ✔ | | |
| SIG | ✔ | ✔ | |
| Dynamic | ✔ | ✔ | |
| TacT | ✔ | | |
| WaNet | | ✔ | |
| SSBA | | ✔ | |
| Adaptive-Blend | ✔ | | |
| Adaptive-Patch | ✔ | | |
| SOS | | | ✔ |
| LWP | | | ✔ |

### B.3 On the selection of Hyperparameters

Within this section, we provide an outline for the selection of the shrinkage parameter $\beta$ utilized in the Shrunk-Covariance Mahalanobis. The primary motivation behind employing SCM is to address the issue of unstable estimation in the inverse of the sample covariance matrix, which can result in non-IID property of samples within $\mathcal{D}^{\text{Val}}$ and future test data. Such non-IIDness can override the order information for those samples, affecting the FPR as described in Theorem 1.

To ensure the IID property of the samples within $\mathcal{D}^{\text{Val}}$ as well as future test samples, we propose the following approach for selecting the shrinkage parameter $\beta$. Firstly, we partition $\mathcal{D}^{\text{Val}}$ into two mutually exclusive datasets, namely $\mathcal{D}_1$ and $\mathcal{D}_2$, with the size of $\mathcal{D}_1$ greater than that of $\mathcal{D}_2$. Next, we compute the Shrunk-Covariance Mahalanobis (SCM) scores based on $\mathcal{D}_1$ and perform a search for an appropriate value of $\hat{\beta}$ that ensures the SCM scores, $s_{\hat{\beta}}(\mathcal{D}_1)$, for the samples within $\mathcal{D}_1$ and $s_{\hat{\beta}}(\mathcal{D}_2)$ are IID samples. To verify the IID property, we conduct a Two-sample Kolmogorov-Smirnov Test

Table 3: Open-source packages used for the results in the *main text*

| | Backdoor ToolBox [75] | BackdoorBench [76] | OpenBackdoor [49] |
|---|:---:|:---:|:---:|
| BadNets | ✔ | | |
| Blended | ✔ | | |
| TrojanNN | ✔ | | |
| SIG | ✔ | | |
| Dynamic | ✔ | | |
| TacT | ✔ | | |
| WaNet | | ✔ | |
| SSBA | | ✔ | |
| Adaptive-Blend | ✔ | | |
| Adaptive-Patch | ✔ | | |
| SOS | | | ✔ |
| LWP | | | ✔ |

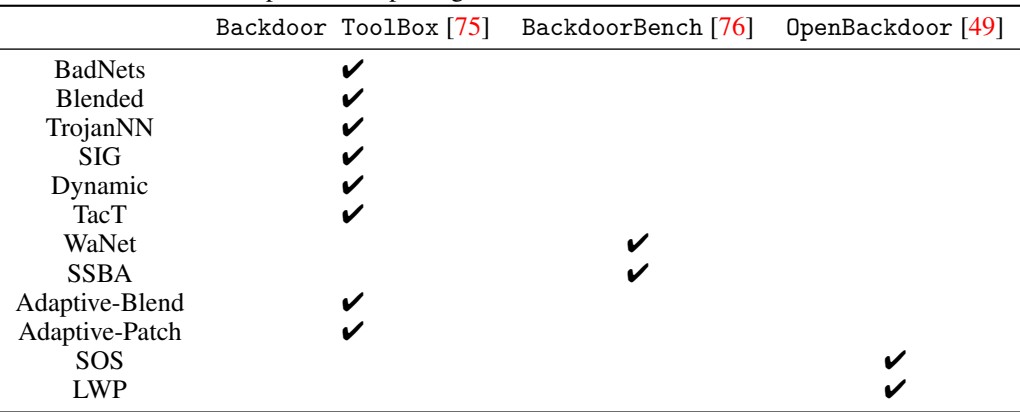

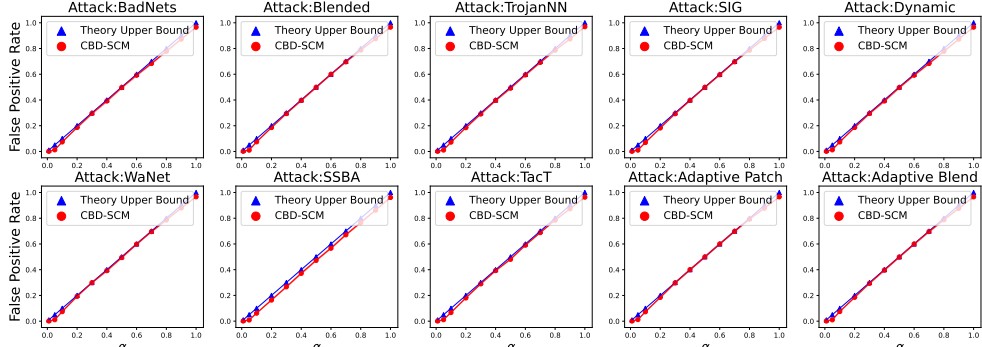

Figure 6: The mean FPRs of our proposed method on CIFAR10 are shown in each plot, which is independently replicated 10 times. The solid line represents the mean value, and the standard errors are < 0.01 for all cases. Our method's FPRs consistently match the theoretical upper bounds.

with a decision rule based on a p-value of 0.05. We repeat the above procedure ten times and observe that for CV (NLP) attacks, $\hat{\beta} = 0.5$ (0.7) satisfies the aforementioned requirement. Consequently, we set $\beta = 0.5$ for all CIFAR10, GTSRB experiments (for Tiny ImageNet: $\beta = 0.1$), and $\hat{\beta} = 0.7$ for all NLP experiments. Ablation studies investigating different values of $\beta$ are included in Appendix F.

## C  Omitted Experimental Results in Main Text

### C.1  Results on the FPR for CIFAR10

This section presents the results of the False Positive Rate (FPR) analysis conducted on CIFAR10. Figure 6 illustrates the performance of our proposed CBD-SCM. Notably, our method consistently achieves FPR values that align with the theoretical upper bounds.

### C.2  Results on the detection power (ROC) for IMDB Dataset

We assess the detection performance of our proposed method on IMDB (NLP) using the base uncased-BERT model. Specifically, we generate ROC curves for our methods under two advanced backdoor attacks, as illustrated in Figure 7. The results indicate that our method outperforms **all** other methods.

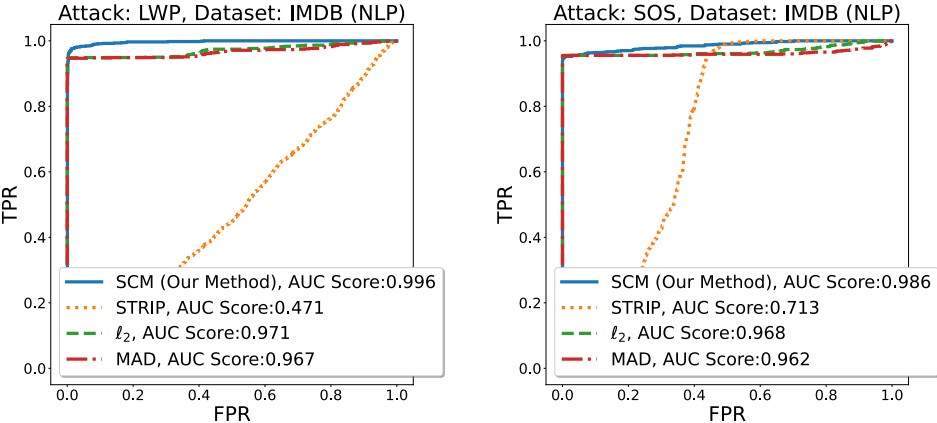

Figure 7: ROC of our SCM method on IMDB NLP. Our proposed method consistently outperforms all other methods.

# D Ablation Study: Different model architectures

In this section, we demonstrate the performance of our proposed methods using the VGG 19 model. The dimension of the `avgpool` layer in VGG 19 is $25088$, which is impractically large. Therefore, we utilize the classifier layer with a dimension of $4096$ for practical purposes. We have observed that the empirical FPR performance of our method consistently aligns with the theoretical bounds in all cases. Consequently, we omit the results for FPR performance and focus on reporting the results for detection power (AUCROC, ROC) in the following subsections.

## D.1 Detection Power on CIFAR10

We assess the detection performance of our proposed method on CIFAR10 using VGG 19. We present the ROC curves for our method in Figure 8 and 9. It is evident that our SCM method consistently outperforms other methods in **all** attack scenarios. Specifically, for certain advanced attacks such as the Dynamic and WaNet, we observe a remarkable improvement of approximately $200\%$ in terms of AUCROC.

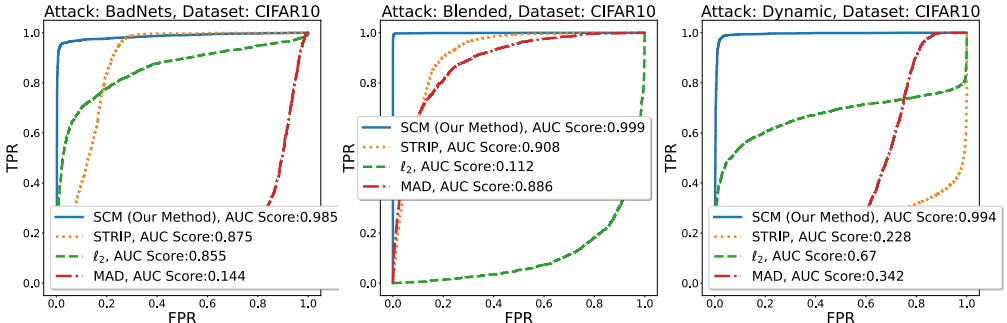

Figure 8: ROC of our method on CIFAR10 with VGG19. Our proposed method consistently outperforms other methods.

## D.2 Detection Power on GTSRB

We evaluate the detection performance of our proposed method on GTSRB using the VGG 19 model. The ROC curves for our method are displayed in Figure 10 and 11. It is evident from the results that our SCM method consistently outperforms other methods in the majority of attack scenarios. Notably, for advanced attacks like SSBA, we observe a significant improvement in terms of AUCROC.

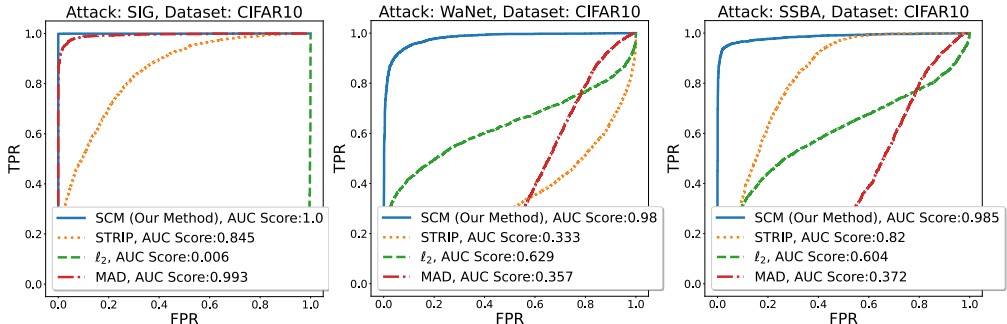

Figure 9: ROC of our method on CIFAR10 with VGG19. Our proposed method consistently outperforms other methods.

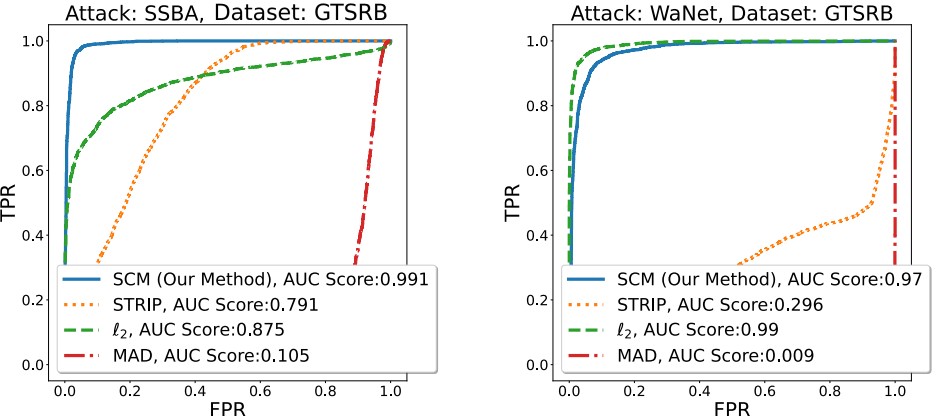

Figure 10: ROC of our method on GTSRB with VGG19.

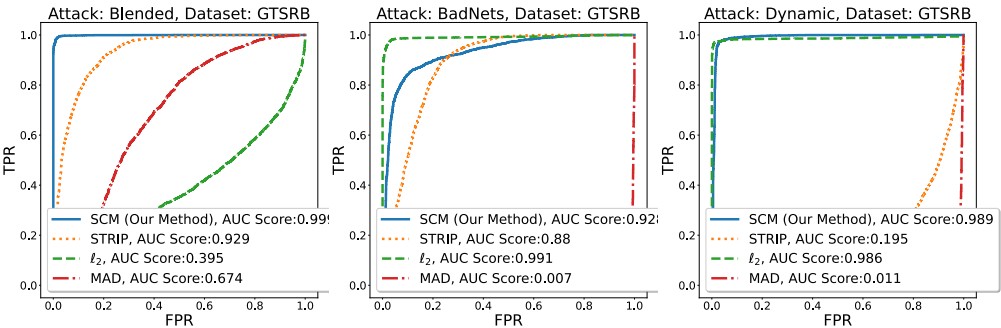

Figure 11: ROC of our method on GTSRB with VGG19.

## E  Ablation Study: Different poisoning ratios

In this section, we provide the results obtained by using different poisoning ratios. The summary of the results for CIFAR10 can be found in Table 4. It is observed that the detection performance of our method exhibits minimal variations across different poisoning ratios. This finding further reinforces the robustness and universal effectiveness of our method.

## F  Ablation Study: Different choices of $\beta$

In this section, we provide empirical studies on the effect of using different $\beta$. Specifically, we select $\beta \in \{0.3, 0.4, 0.6, 0.7\}$.

Table 4: AUCROC score of our method on CIFAR10

| Poisoning Ratio → | 0.3% | 1% | 5% |
|---|---|---|---|
| BadNets | 0.99 | 0.99 | 0.99 |
| Blended | 0.96 | 0.96 | 0.96 |
| WaNet | 0.88 | 0.91 | 0.94 |
| SSBA | 0.92 | 0.95 | 0.97 |

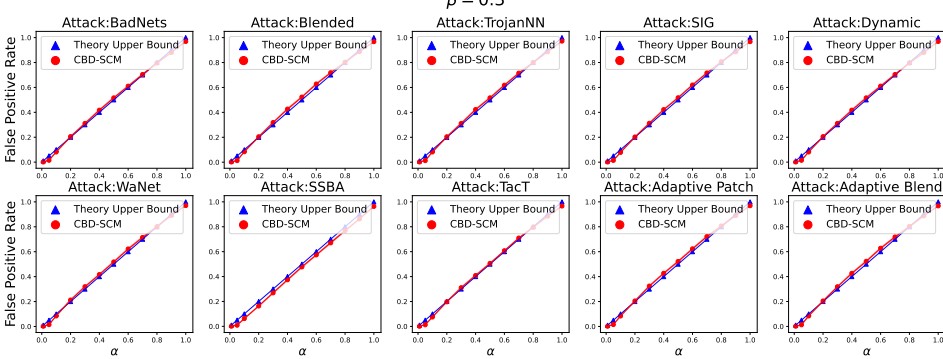

Figure 12: The mean FPRs of our proposed method with $\beta = 0.3$ on CIFAR10 are shown in each plot, which is independently replicated 10 times. The solid line represents the mean value, and the standard errors are $< 0.01$ for all cases.

## F.1 Performances on FPR for CIFAR10

This section presents the results of the False Positive Rate (FPR) analysis conducted on CIFAR10 with different choices over $\beta$. Figure 12, 13, 14, 15 illustrate the FPR performance of our proposed CBD-SCM, with $\beta = 0.3, 0.4, 0.6, 0.7$ respectively. Our method consistently achieves FPR values that align with the theoretical upper bounds.

## F.2 Performances on detection power (AUCROC, ROC) for CIFAR10

We assess the detection performance of our proposed method on CIFAR10 with different choices over $\beta$. Figure 16, 17, 18, 19 illustrate the ROC of our proposed SCM, with $\beta = 0.3, 0.4, 0.6, 0.7$ respectively. The AUCROC scores exhibit minimal variations when different values of $\beta$ are used, indicating the consistent and robust effectiveness of our method. This observation reinforces the stability and universality of our selected $\beta = 0.5$ as reported in the main text.

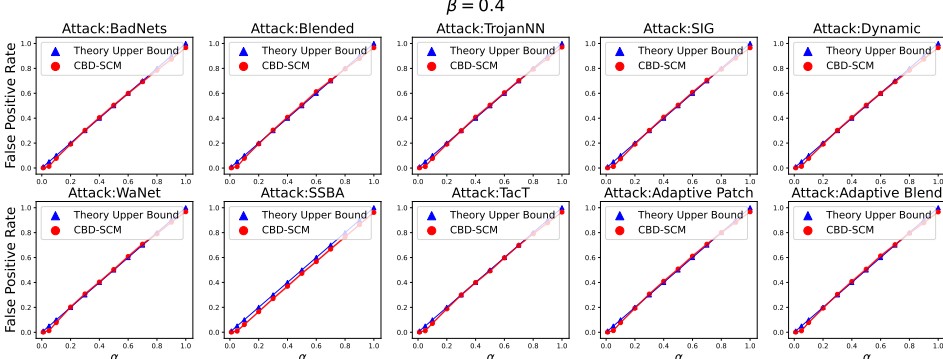

Figure 13: The mean FPRs of our proposed method with $\beta = 0.4$ on CIFAR10 are shown in each plot, which is independently replicated 10 times. The solid line represents the mean value, and the standard errors are $< 0.01$ for all cases.

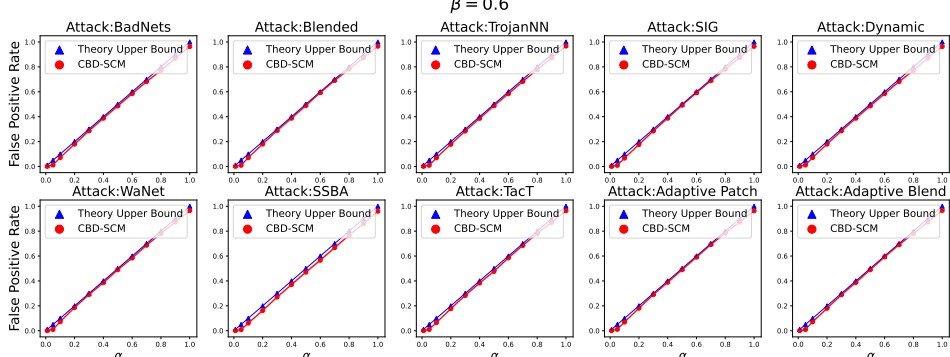

Figure 14: The mean FPRs of our proposed method with $\beta = 0.6$ on CIFAR10 are shown in each plot, which is independently replicated 10 times. The solid line represents the mean value, and the standard errors are < 0.01 for all cases.

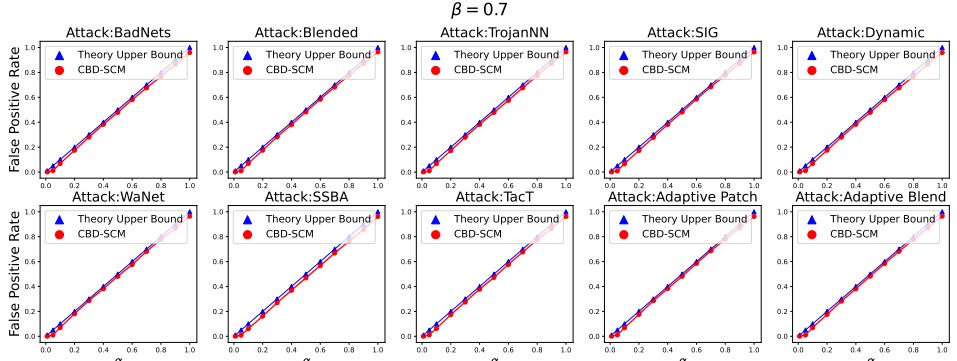

Figure 15: The mean FPRs of our proposed method with $\beta = 0.7$ on CIFAR10 are shown in each plot, which is independently replicated 10 times. The solid line represents the mean value, and the standard errors are < 0.01 for all cases.

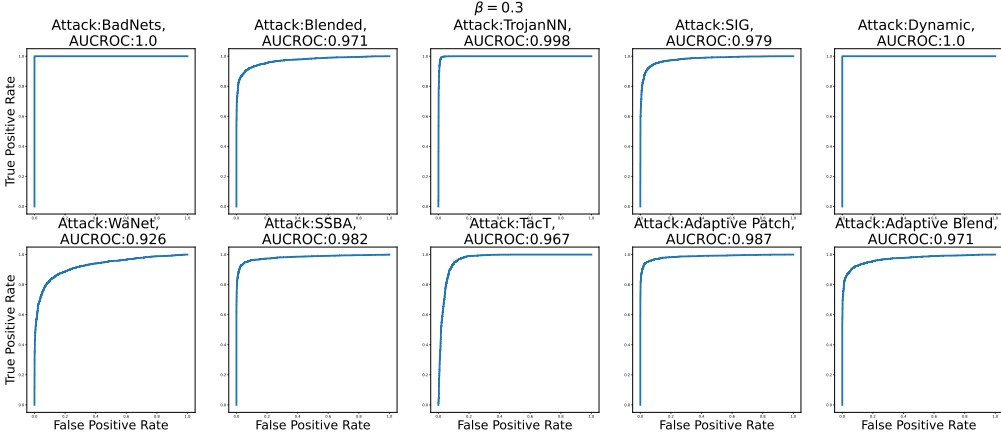

Figure 16: ROC of our method on CIFAR10 with $\beta = 0.3$

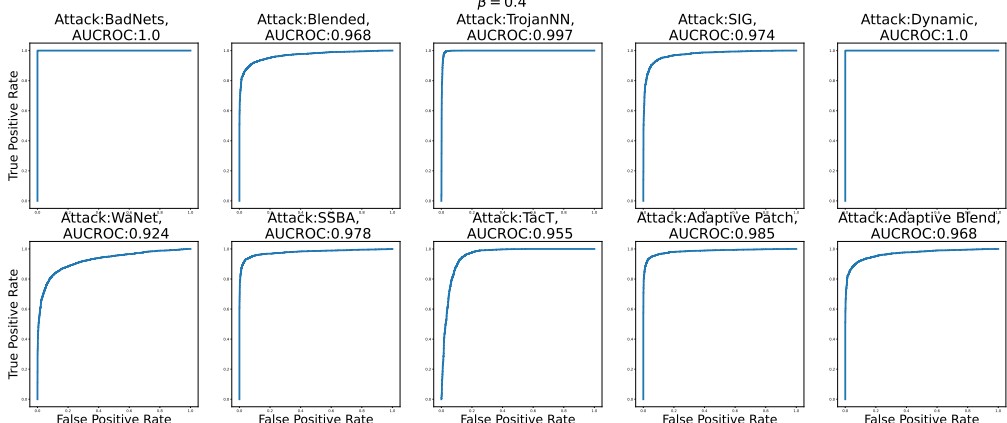

Figure 17: ROC of our method on CIFAR10 with $\beta = 0.4$

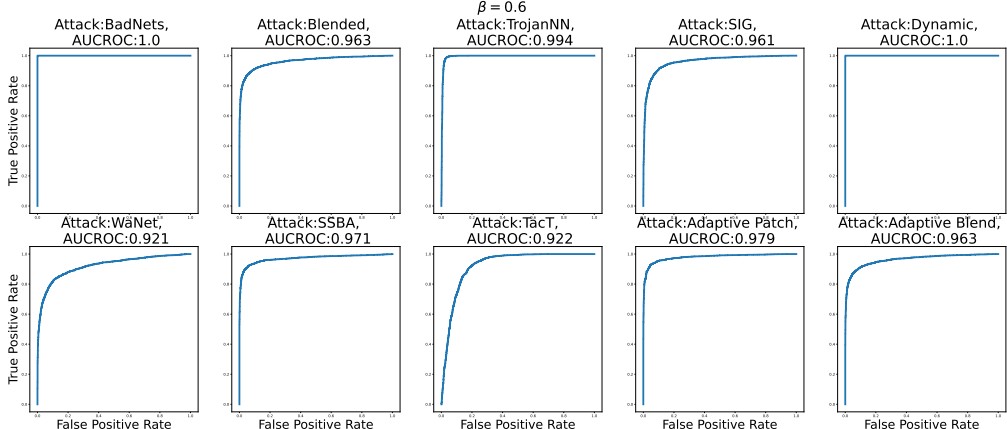

Figure 18: ROC of our method on CIFAR10 with $\beta = 0.6$

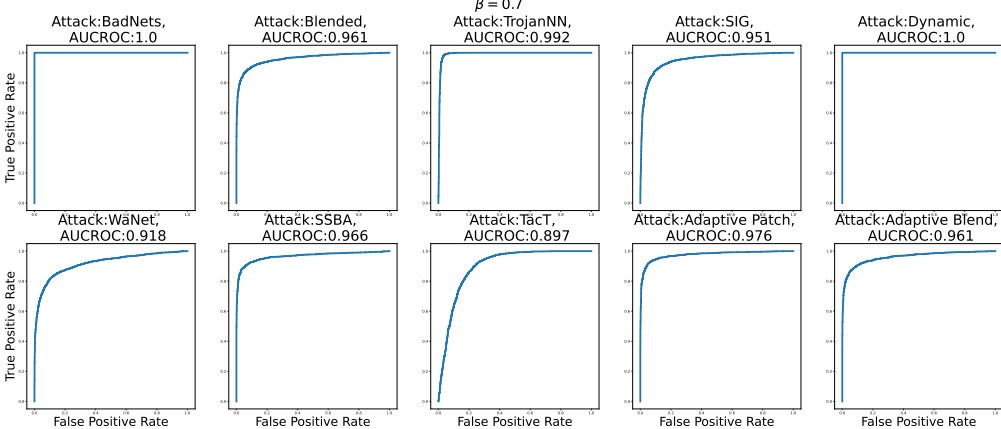

Figure 19: ROC of our method on CIFAR10 with $\beta = 0.7$

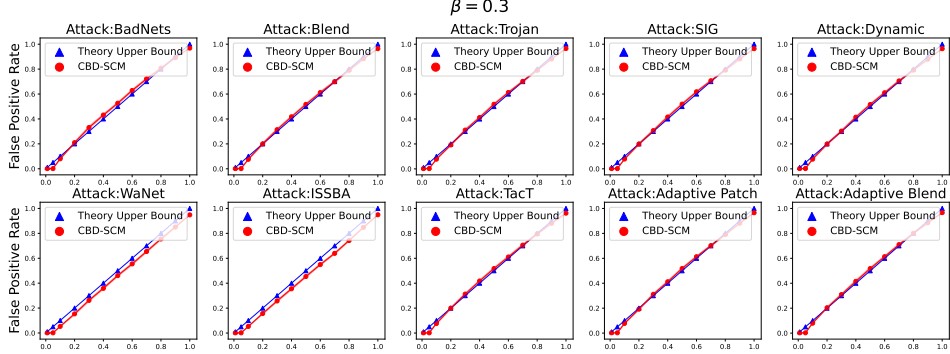

Figure 20: The mean FPRs of our proposed method with $\beta = 0.3$ on GTSRB are shown in each plot, which is independently replicated 10 times. The solid line represents the mean value, and the standard errors are < 0.01 for all cases.

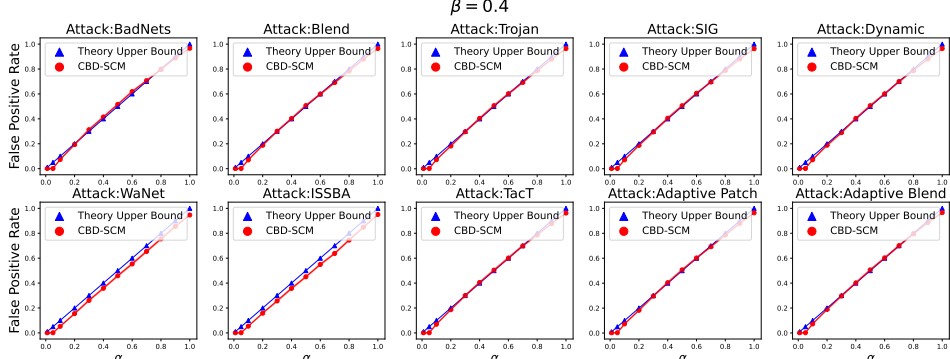

Figure 21: The mean FPRs of our proposed method with $\beta = 0.4$ on GTSRB are shown in each plot, which is independently replicated 10 times. The solid line represents the mean value, and the standard errors are < 0.01 for all cases.

### F.3 Performances on FPR for GTSRB

This section presents the results of the False Positive Rate (FPR) analysis conducted on GTSRB with different choices over $\beta$. Figure 20, 21, 22, 23 illustrate the FPR performance of our proposed CBD-SCM, with $\beta = 0.3, 0.4, 0.6, 0.7$ respectively. Our method consistently achieves FPR values that align with the theoretical upper bounds.

### F.4 Performances on detection power (AUCROC, ROC) for GTSRB

We assess the detection performance of our proposed method on GTSRB with different choices over $\beta$. Figure 24, 25, 26 illustrate the ROC of our proposed SCM, with $\beta = 0.3, 0.4, 0.6$ respectively. The AUCROC scores exhibit minimal variations when different values of $\beta$ are used, indicating the consistent and robust effectiveness of our method. This observation reinforces the stability and universality of our selected $\beta = 0.5$ as reported in the main text.

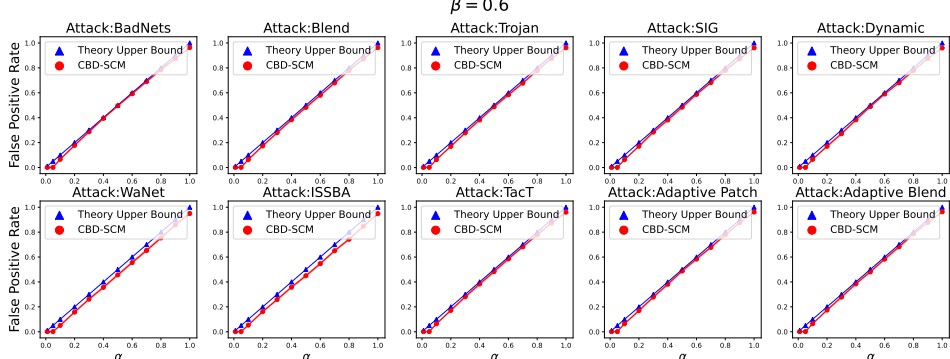

Figure 22: The mean FPRs of our proposed method with $\beta = 0.6$ on GTSRB are shown in each plot, which is independently replicated 10 times. The solid line represents the mean value, and the standard errors are < 0.01 for all cases.

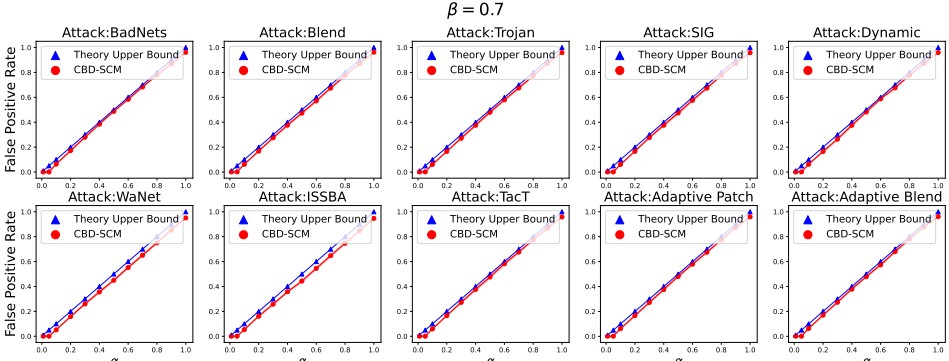

Figure 23: The mean FPRs of our proposed method with $\beta = 0.7$ on GTSRB are shown in each plot, which is independently replicated 10 times. The solid line represents the mean value, and the standard errors are < 0.01 for all cases.

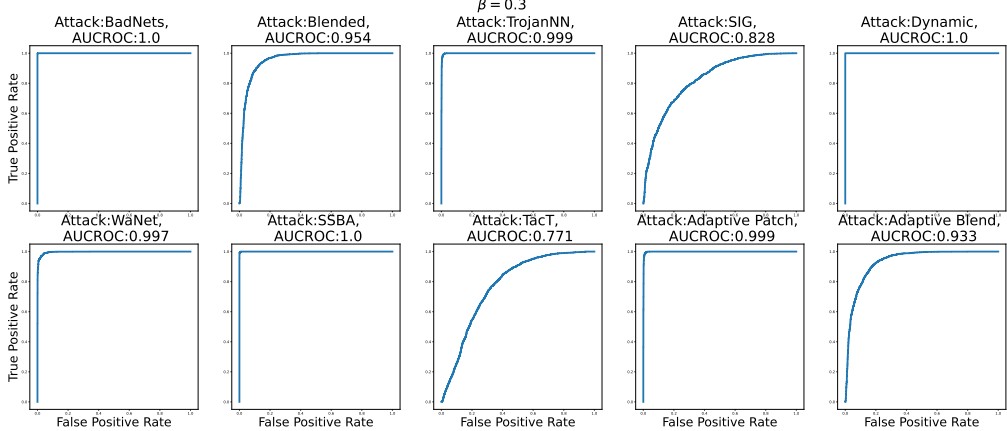

Figure 24: ROC of our method on GTSRB with $\beta = 0.3$

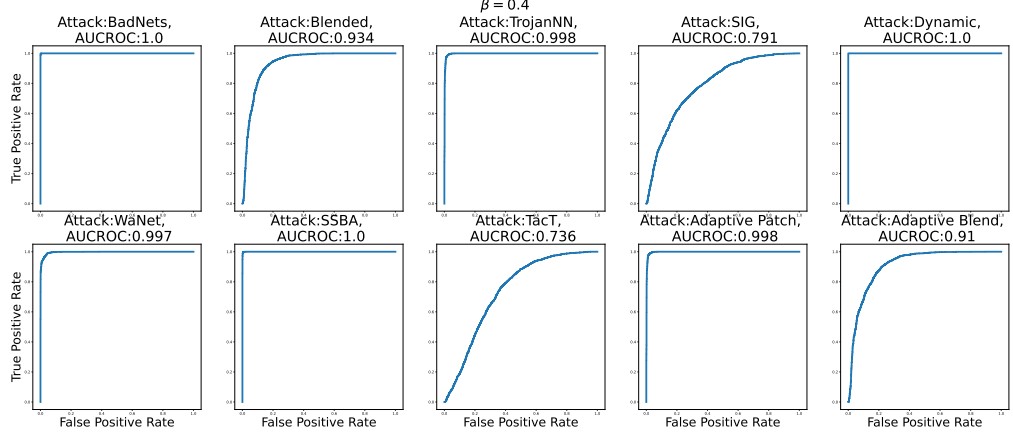

Figure 25: ROC of our method on GTSRB with $\beta = 0.4$

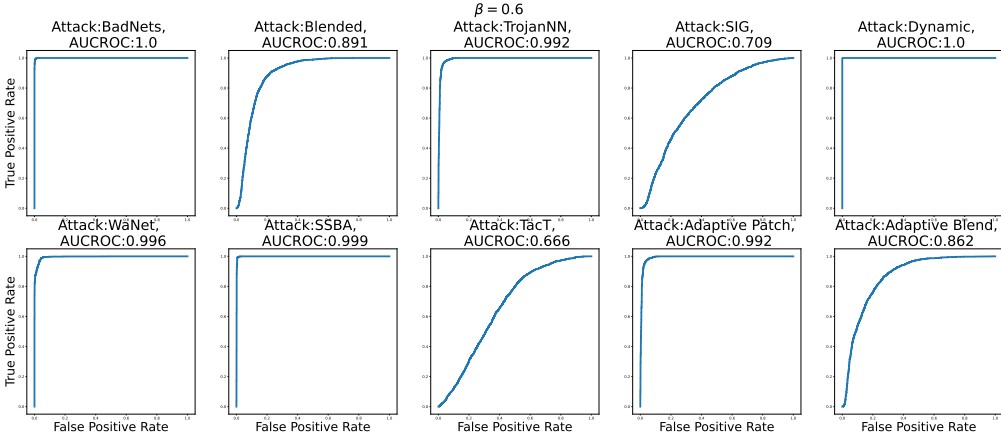

Figure 26: ROC of our method on GTSRB with $\beta = 0.6$

## G  Performances under additional types of backdoor attacks

We evaluate our methods against backdoor attacks intentionally designed to challenge our defenses, such as LIRA [12] and M-attack (a custom-designed attack). These attacks are specifically crafted to diminish the distinction in latent spaces of the backdoored models by imposing regularization on the distances. LIRA employs Wasserstein distances, while M-attack utilizes Mahalanobis distances, to measure the dissimilarity between clean data and backdoor attacks. In Table 5, our methods consistently outperform state-of-the-art defenses, although there is a slight performance dip compared to diverse attack scenarios like BadNets and SSBA. Nevertheless, this outcome is reasonable since no defense can be universally effective against all attack variations.

Table 5: AUCROC score of our method on GTSRB against LIRA and M-attack.

| Defenses → | SCM (Ours) | FREQ | SCALEUP |
|---|---|---|---|
| LIRA | **0.86** | 0.71 | 0.79 |
| M-attack | **0.82** | 0.80 | 0.71 |

## H  Comparison with other defenses

### H.1  Performances comparison with recent detection-based backdoor defenses

We evaluate our methods alongside two recently developed detection-based defenses designed to counteract CV backdoor attacks: SCALEUP and FREQ. The summarized results are presented in

Table 6 below. Our observations consistently demonstrate that our methods outperform SCALEUP and FREQ.

Table 6: AUCROC score of our method on GTSRB. The bestperforming method(s) are indicated in boldface.

| Defenses → | SCM (Ours) | FREQ | SCALEUP | ABL | SPECTRE |
|------------|------------|------|---------|-----|---------|
| BadNets | **0.99** | 0.91 | 0.86 | 0.97 | 0.96 |
| SSBA | **0.99** | 0.51 | 0.72 | 0.81 | 0.56 |
| Adaptive Patch | **0.87** | 0.49 | 0.55 | 0.72 | 0.70 |
| Adaptive Blend | **0.99** | 0.56 | 0.51 | 0.59 | 0.62 |

## H.2 Performances comparison with purifying-based training-stage backdoor defenses

We assess our methods in conjunction with two training-stage backdoor defenses, SPECTRE and ABL. As emphasized in the main text, these methods, while sharing the concept of distinguishing between clean and backdoor attacks, significantly differ from ours in terms of the threat model, methodology, and evaluation metrics. To ensure a fair comparison, we report the AUCROC scores for distinguishing between clean and backdoor data. The summarized results are presented in Table 6. Our observations consistently demonstrate that our methods outperform ABL and SPECTRE.

