# OpenReview forum: "A Unified Detection Framework for Inference-Stage Backdoor Defenses"
_NeurIPS.cc/2023/Conference — NeurIPS 2023 poster_

### Official Review · Reviewer_1Ekh · 2023-07-03

**Soundness:** 2 fair
**Presentation:** 3 good
**Contribution:** 2 fair
**Rating:** 5
**Confidence:** 4

**Summary:**

This work formulates the inference-stage backdoor detection problem. The authors then propose a framework to establish provable guarantees w.r.t. the detection FPR, given some validation data on hand. Finally, they derive the optimal detection rule (in the Neyman-Pearson paradigm) in a simplified scenario, and suggest a practical proxy using the empirical Mahalanobis distance metric w.r.t. DNN's latent representations. Their results on both CV and NLP domain show significant improvements over prior arts

**Strengths:**

1. The authors formally study the inference-stage backdoor detection problem. Specifically, they propose the conformal backdoor detection (CBD) framework, which ensures the detector's FPR does not deviate too much from a pre-selected value with a high probability. Their CBD framework provides theoretical guidance on how to select the decision threshold $\tau$.
2. They also study the optimal score function in a simplified scenario. Above this theretical analysis, they propose a practical proxy to this score function (which cannot be practically computed). They further address the potential numerical instability problem using matrix shrinkage technique.
3. The authors conduct extensive experiments, in both CV and NLP domain. The cross-modal experiment setup indeed helps demonstrate the generalizability of their method. They consider 3 datasets and 10 attacks for CV tasks, with 2 datasets and 2 attacks for NLP task. I appreciate the diverse experimental setting very much.

**Weaknesses:**

1. The paper title is not expressing your work accruately. Your title is "A Unified Framework for Inference-Stage Backdoor **Defenses**", but your work focuses only on inference-stage backdoor **input detection** (I don't think inference-stage backdoor defense = inference-stage backdoor input detection). Also, the word "unified" seems quite strong, since your theoretical analysis (e.g. optimality) is mostly based on a simplified scenario. I suggest you narrow down the scope (and consider tuning down the tone) of your title.
2. The **intuitions behind of your method** are not fully specified. For example, as a reader, I am quite confused about why the empirical Mahalanobis distance (Line 276) is a good proxy to Eq (2)? It seems like the case that you simply discard the second term in Eq (2), and try to only approximate the first term. Please explain more about the connection between your practical proxy and Eq (2).
3. The experiment setting should include comparisons with **more baselines defenses and attacks**. First, I suggest the authors also consider other non-poisoning backdoor attacks (e.g. modifying trained model weights [1], fine-tuning, etc.). Second, there are more recent and advanced inference-stage backdoor detector baselines (e.g. [2] and [3]) other than STRIP, l2 and MAD. You should definitely consider adding at least one or two inference-stage backdoor detectors in the recent two years into comparison in the main Table 1.
4. It's good to see several ablation studies in the Appendix (e.g. poison rates). I suggest you consider three more important **ablation studies**: 1) The number of validation data you have in hand. How many samples at least are necessary to make your defense effective? 2) I appreciate your consideration of several adpative attacks. But these adaptive attacks are "adaptive" against the latent-separation based defenses. Could you also study/propose potential adaptive attacks that specifically target the weakness of your method? 3) What if the validation data is OOD (e.g. corrupted with noises). In the real world, model deployers sometimes cannot guarantee to have the exactly validation data drawn IID from the training samples' distribution.
5. **Typos**: Line 147 "to to"; Line 214 "at lease"; Line 230 "we will assume that the Suppose that"; Line 264 $\eta^* \to \gamma^*$ (?).

Still, I would be happy to adjust my rating if the above concerns are somehow addressed.

[1] Qi, Xiangyu, Tinghao Xie, Ruizhe Pan, Jifeng Zhu, Yong Yang, and Kai Bu. "Towards practical deployment-stage backdoor attack on deep neural networks." In *Proceedings of the IEEE/CVF Conference on Computer Vision and Pattern Recognition*, pp. 13347-13357. 2022.

[2] Zeng, Yi, Won Park, Z. Morley Mao, and Ruoxi Jia. "Rethinking the backdoor attacks' triggers: A frequency perspective." In *Proceedings of the IEEE/CVF international conference on computer vision*, pp. 16473-16481. 2021.

[3] Guo, Junfeng, Yiming Li, Xun Chen, Hanqing Guo, Lichao Sun, and Cong Liu. "Scale-up: An efficient black-box input-level backdoor detection via analyzing scaled prediction consistency." *arXiv preprint arXiv:2302.03251* (2023).

**Questions:**

1. In Line 109-111, you made a claim that "these methods rely on the assumption that there is a clear separation...". But isn't your method also relying on the separation of clean and backdoor data in the latent space? Do you consider utilizing such latent separation as a weakness/flaw when designing backdoor defenses? If you do, why won't your method suffer from this weakness?
2. How is the violation rate $\delta$ in Eq (1) selected in practice?
3. In Figure 2, I see that for some attacks (TrojanNN and A-Blend), even your proposed optimal scoring function cannot separate well enough between clean and backdoor data. Could you discuss more about these results?
4. Could youn also visualize the SCM score histogram for backdoor and clean inputs? It would be straightforward for readers to see how your proposed method distinguishes between clean and backdoor inputs.

**Limitations:**

The limitations are discussed together with future work in Sec 5.

---

> ### Author Rebuttal · Authors · 2023-08-09
>
> > Q1: Inaccurate terms in the title
>
> R: We appreciate your suggestion to improve the current title to better align with specific content, such as focusing on inference-stage backdoor input detection. Your input is valuable, and we will carefully incorporate your suggestions in our revision process.
>
> > Q2: Intuitions behind the proxy term
>
> R: Recall that the optimal decision rule is represented as $s(Z) \propto P_{\text{BD}}(Z)/P_{\text{CL}}(Z)$, where $P_{\text{BD}}(\cdot)/P_{\text{CL}}(\cdot)$ denotes the distribution of backdoor and clean data, respectively.
> Since we do not have access to the backdoor data, computing $P_{\text{BD}}(\cdot)$ is not possible.
> As an alternative, empirically, we consistently observed that $P_{\text{CL}}(\text{Clean Data})$ is much greater than $P_{\text{CL}}(\text{Backdoor Data})$. Hence, we use $P_{\text{CL}}(\cdot)$ as a substitute for $s(Z)$. When considering Gaussian data assumptions, the proxy $P_{\text{CL}}(Z) \propto 1/\exp{(0.5Z^{\top}\Sigma Z)}$ effectively represents the inverse of the Mahalanobis distance to the clean data distribution, which explains the use of SCM as the proxy.
>
> > Q3:  Comparison with more baseline defenses [2], [3] and attack [1]
>
> R: We conducted two sets of experiments: (1) assessing our method against [1], and (2) comparing our method with [2] and [3]. We present the outcomes in **Table 2 in the PDF file of the global response**, respectively. In summary, our methods excel in non-poisoning backdoor attacks [1]. Also, they consistently outperform [2] and [3].
>
> > Q4: Claim in Lines 109 - 111
>
> Yes, our methods also assume a separation between the latent representations of clean and backdoor data.
> Nevertheless, our approach diverges from other work mentioned in the Related Work section in how we exploit this distinction property. We utilize SCM, consistently outperforming previous methods.
>
> However, an issue arises if knowledgeable attackers know the SCM score and exploit it to bypass our defense, a concern we'll address next.
>
>
>
>
> > Q5: Potential adaptive attacks that specifically target the weakness of our method
>
> R: Following the previous response, we developed a new attack, referred to as *M-attack*, which introduced a regularization term to reduce the Mahalanobis scores between clean and backdoor samples. Experiment outcomes are presented in **Table 2 in the PDF file of the global response**. Our methods maintain superiority over other defenses, with a moderate performance decrease compared to different attacks like BadNets and SSBA. This is reasonable, considering the nature of the newly proposed attack.
>
>
> > Q6: The selection of $\delta$
>
> R: Choosing the value of $\delta$ should be contingent on both $n$ (sample size) and $\alpha$ (desired type I error rate). This ensures the establishment of a testing procedure that holds a high probability guarantee on the desired type I error rate.
>
> To be specific, it can be shown that, with similar arguments in Theorem 1, to ensure a $1-\delta$ probability guarantee on the type I error rate of $\alpha$, the values of $n$, $\delta$, and $\alpha$ must lead to the right-hand side term of Eq. (1), i.e.,
> $1-\alpha+\sqrt{\log(2/\delta)/(2n)} $ being strictly less than 1.
> For instance, consider the values $n=200$, $\alpha = 0.05$, and $\delta = 0.01$. In this case, we have $ 1-0.05+\sqrt{\log(2/0.01)/400} > 1,$ indicating the infeasibility of achieving a type I error rate less than $\alpha = 0.05$ with a probability of at least $1 - 0.01$.
>
>
> > Q7: Ablation studies on the size of the validation data and OOD validation datasets.
>
> R: Firstly, we'd like to draw your attention to the **challenges** linked with establishing a provable type I error rate in the two scenarios you mentioned. As we highlighted in our previous response, a validation size of 200 doesn't offer a high probability guarantee for the type I error rate. Moreover, when using out-of-distribution (OOD) data as the validation dataset, it can generally be shown that ensuring any guarantees for the type I error rate becomes **unfeasible**. Hence, these two scenarios might not be the primary application contexts for our methods, which primarily emphasize defense techniques with provable type I error guarantees and strong detection power.
>
> Nevertheless, according to your suggestions, we conducted two ablation studies involving variations in the validation dataset size and the inclusion of OOD data in the clean validation dataset. The summarized results are presented in **Table 3, and 4 respectively in the PDF file of the global response**. Our observations indicate that our methods are consistently effective when the validation dataset size is above 200. Moreover, if the validation dataset consists of less than 25% of the OOD samples, our methods exhibit consistent effectiveness. These findings highlight the robustness of our approaches.
>
> > Q8: For some attacks (TrojanNN and A-Blend), even your proposed optimal scoring function cannot separate well enough between clean and backdoor data.
>
> R: We provide an explanation for the less distinct separation between clean and backdoor data under optimal score rules as follows.
> It's important to remember that the uniformly most powerful rule in Eq.(2) is established based on various conditions, including the presence of fully specified Gaussian data distributions for both clean and backdoor data. However, our investigations revealed a significant departure from Gaussian distributions in the latent spaces of clean and backdoor data during TrojanNN and A-Blend attacks. This divergence leads to a violation of the assumptions necessary for the application of the optimal decision rule in Eq.(2).
>
> > Q9: Histograms of SCM scores
>
> R: We have incorporated the SCM score histograms in **Figure 2 the global response PDF file**. In summary, we notice distinct separations between the SCM scores of clean and backdoor samples consistently across various scenarios.

---

> > ### Comment · Reviewer_1Ekh · 2023-08-16
> > **Thanks for your effort and clarifications**
> >
> > I appreciate your efforts in the rebuttal very much! I do believe most of my concerns are resolved. A few suggestions / further concerns:
> > - Your "uniformly most powerful rule" is established in a simplified cases where you assume "Gaussian data distributions for both clean and backdoor data". Therefore, your defense may not generalize well to attacks which violate such assumptions or to scenarios when the defender cannot acquire enough validation clean samples. Still, I appreciate your specifically designed adaptive attack & analysis a lot. Then please tone down your statement like "uniformly most powerful rule".
> > - I also noticed the concern raised by reviewer ibgE about your work's similarity to SPECTRE. Indeed, I also find it to be beneficial to include an independent section to comprehensively discuss about your work's relationship (especially similarity) to it (and also other latent-space analysis backdoor defense) in the major paper (since you are both conducting an outlier analysis in the latent representation space).
> > - Also, in Table 2 of your rebuttal PDF, why is Frequency performing so bad on SSBA? Could you elaborate the experiment configurations, e.g. whether you are using their pretrained models? According to my own experiment experience, Frequency can actually achieve a much higher AUCROC against SSBA on CIFAR10.
> >
> > Thanks again for your efforts during the rebuttal period. I will adjust my rating to 5 for now.

---

> > > ### Author Response · Authors · 2023-08-16
> > > **Thanks Reviewer 1Ekh's feedback on our rebuttal and increasing the score; Further Clarifications**
> > >
> > > Thank you for your valuable feedback on our rebuttals and for increasing the score. We address your further concerns/suggestions in the following.
> > >
> > > > Q1: Your "uniformly ... Then please tone down your statement like "uniformly most powerful rule".
> > >
> > > R: Per our prior response, we acknowledge the need for a more specific title and moderated language. We promise to revise both the title and content with accuracy and an appropriate tone.
> > >
> > > > Q2:  Include an independent section to comprehensively discuss about your work's relationship (especially similarity) to it (and also other latent-space analysis backdoor defense) in the major paper (since you are both conducting an outlier analysis in the latent representation space).
> > >
> > > R: In the revision, we'll incorporate a dedicated section to thoroughly explore the connection between our approach and other comparable methodologies.
> > >
> > > > Q3:  why is Frequency performing so bad on SSBA? Could you elaborate the experiment configurations, e.g. whether you are using their pretrained models?
> > >
> > > R: Regarding the subpar performance of SSBA on the *GTSRB* dataset presented in *Table 2* of the global response PDF, this outcome aligns with previous observations indicating that the Frequency method fails against advanced non-patch-based backdoor attacks, such as SSBA and WaNet [C]. Specifically, the study by [C], introducing a defense approach at the ICLR 2023 event as you mentioned, reported AUC scores around 0.5 for the Frequency defense method when applied against the SSBA attack on both the CIFAR10 and Tiny ImageNet datasets. These outcomes parallel our findings, corroborating that the Frequency method inadequately addresses the SSBA attack.
> > >
> > > In terms of the implementation, we implemented all the backdoor attacks, including the SSBA attack, by using open-source packages, e.g., [A] and [B]. The resulting backdoor model achieves both high clean and backdoor accuracy on the CIFAR10, the GTSRB, and the Tiny ImageNet dataset. For instance, a clean accuracy of 95\% and a backdoor accuracy of 97\% accuracy are observed on the GTSRB dataset for the SSBA attack.
> > >
> > >
> > > Thank you again for your feedback on our response and the increased score. If you have any more questions, feel free to let us know.
> > >
> > > ### Reference
> > >
> > > [A] T. Xie, “Backdoor toolbox,” https://github.com/vtu81/backdoor-toolbox, 2022.
> > >
> > > [B] B. Wu, H. Chen, M. Zhang, Z. Zhu, S. Wei, D. Yuan, and C. Shen, “Backdoorbench: A comprehensive benchmark of backdoor learning,” in NIPS Datasets and Benchmarks Track, 2022.
> > >
> > > [C] Guo, Junfeng, Yiming Li, Xun Chen, Hanqing Guo, Lichao Sun, and Cong Liu. "Scale-up: An efficient black-box input-level backdoor detection via analyzing scaled prediction consistency." in ICLR 2023.

---

> > > > ### Author Response · Authors · 2023-08-19
> > > >
> > > > We sincerely appreciate Reviewer 1Ekh for their insightful and positive comments. In our response, we have addressed concerns regarding  (1) utilizing appropriate tones for both the title and the main text, and (2) assessing the performance of a method you referenced.
> > > >
> > > > As the discussion phase nears its conclusion, we kindly inquire if the reviewer has any further comments on our response. We are readily available for any additional queries they may have.
> > > >
> > > > Once more, we appreciate your time and effort in reviewing our paper.

---

> > > > > ### Comment · Reviewer_1Ekh · 2023-08-19
> > > > > **Thanks for the response.**
> > > > >
> > > > > I do not have any further comment / response at this moment. Thank the authors again for their great effort during the rebuttal and discussion period to resolve my concerns.

---

> > > > > > ### Author Response · Authors · 2023-08-19
> > > > > > **Thanks the Reviewer 1Ekh**
> > > > > >
> > > > > > Thank you for your helpful feedback on our responses. We'll make sure to include these suggestions when revising our work.

---

### Official Review · Reviewer_ibgE · 2023-07-04

**Soundness:** 2 fair
**Presentation:** 3 good
**Contribution:** 2 fair
**Rating:** 4
**Confidence:** 4

**Summary:**

This paper proposes a backdoor sample detection method. It utilizes Mahalanobis distance as the score function to compute the probability of a given sample being poisoned. It also leverages an existing statistical tool, the conformal prediction framework, to determine a statistical threshold for the computed scores given a false positive rate (recognizing clean data as poisoned). The experiments are conducted on three image datasets and two text datasets. Comparing to several baseline methods, the proposed approach achieves higher detection rate on poisoned samples.

**Strengths:**

1. This paper studies an important problem of detecting backdoor samples. The evaluation shows the proposed approach is effective against various attacks.

2. The conformal prediction framework is used in the paper to statistically balance the true positive rate and the false positive rate, which is interesting. The Mahalanobis distance is leveraged to differentiate the statistical difference between clean and poisoned data, which is empirically validated in the paper.


**Weaknesses:**

1. While this paper compares with a few baselines, it still misses a closely related work SPECTRE [41]. SPECTRE also leverages statistics techniques to estimate the mean and the covariance of the clean data, which is then utilized to differentiate poisoned samples from the clean data. What is the fundamental difference between the proposed approach and SPECTRE? This paper seems to just use a different statistics tool to estimate the clean distribution. Fundamentally, it is no different from SPECTRE. However, there is no discussion, comparison, and empirical evaluation regarding SPECTRE.

2. Although the evaluation considers a number of attacks, an important aspect is missed in the paper. A knowledgeable attacker who is aware of the proposed detection approach can design an attack specifically targeting it. Particularly, this paper uses the Shrunk-Covariance Mahalanobis (SCM) score function to distinguish clean and poisoned data. An adaptive attacker can use this function during attack to reduce the scores for poisoned samples. This is critical to demonstrate the robustness of the proposed detection approach. In addition, there are several strong attacks that were designed to evade statistics based defenses [1,2]. They should be empirical evaluated.

3. In Algorithm 1, a transformation method T is introduced as the defender-constructed transformation on the original input. However, there is no explanation on what this function is and how it is applied on the input.


[1] Doan, Khoa, Yingjie Lao, and Ping Li. "Backdoor attack with imperceptible input and latent modification." Advances in Neural Information Processing Systems 34 (2021): 18944-18957.

[2] Shokri, Reza. "Bypassing backdoor detection algorithms in deep learning." 2020 IEEE European Symposium on Security and Privacy (EuroS&P). IEEE, 2020.

**Questions:**

See above.

---

> ### Author Rebuttal · Authors · 2023-08-09
>
> We sincerely thank reviewers for investing their time and energy into reviewing our manuscript and offering valuable feedback. We're pleased that they recognized the quality of our writing, acknowledged the novelty of our proposed framework (Reviewer vAAp), and found our approach effective across various domains and setups (JEgp, ibgE, 1Ekh). We will incorporate all their comments into the revised paper.
>
> > Q:  Comparaison with SPECTRE [41]
>
> R: Fundamentally, our approach diverges significantly from SPECTRE in terms of both the threat model and technical components, detailed as follows.
>
> 1. **Differences in terms of the threat model, goals, and evaluation metrics**. SPECTRE [41] is a training-stage defense approach with access to both clean and backdoor training data, **affording them information about both types of clean and backdoor data**. Their goal is to differentiate between clean and backdoor training data, ultimately constructing a clean model through the use of filtered clean training data, obtained by analyzing all training instances. Hence, their evaluation metrics are the clean and attack success rate of the cleansed model.
> In contrast, our method relies solely on a small (e.g., 1000) set of clean validation data and operates **without any knowledge of backdoor data**. Our goal is to detect future backdoor inputs, and our evaluation metric is the AUCROC score of the detector.
>
> 2. **Differences in terms of Technical Aspects**. SPECTRE employs robust statistics to separate clean and backdoor training data, with **access to both clean and backdoor training data**. On the other hand, our SCM technique is employed to mitigate numerical challenges when estimating high-dimensional covariance matrices for the set of clean validation data. Notably, there is **no information available regarding the backdoor data distributions**.
>
>
> Therefore, direct comparisons between SPECTRE [41] and our methods may not be equitable or appropriate.
>
> **Nonetheless**, we have included the AUCROC scores of SPECTRE on distinguishing between clean and backdoor data in Tables 1 and 2 below. We observed that our method consistently outperforms SPECTRE under different types of backdoor attacks. These results underscore our method's superiority over SPECTRE in terms of detection performance.
>
> Table 1: AUCROC performance comparison of our method with SPECTRE [41] on CIFAR10
> | Defense ↓        | BadNets| Dynamic | SSBA | Adaptive-Blend | Adaptive-Patch |
> | :---------------- | :------: | :----: |:----: |:----: |:----: |
> | Our Method       |   1.0  | 1.0 | 0.97 | 0.96 | 0.98 |
> |   SPECTRE [41]  |  0.95 | 0.96 | 0.56 | 0.62 | 0.61|
>
> Table 2: AUCROC performance comparison of our method with SPECTRE [41] on GTSRB
> | Defense ↓        | BadNets| Dynamic | SSBA | Adaptive-Blend | Adaptive-Patch |
> | :---------------- | :------: | :----: |:----: |:----: |:----: |
> | Our Method       |   0.99  | 0.99 | 0.99 | 0.99 | 0.87 |
> |   SPECTRE [41]  |  0.96 | 0.95 | 0.56 | 0.62 | 0.7|
>
> > Q: Robustness of the proposed detection approach against a knowledgeable; comparison with strong attacks that were designed to evade statistics-based defenses [1,2].
>
> R: Taking your advice into account, we developed an attack inspired by [1], refered to as *M-attack*, which introduced a regularization term to reduce the  Mahalanobis scores between clean and backdoor samples. Moreover, in response to your recommendations, we conducted supplementary experiments to evaluate our method's effectiveness against attacks [1,2], as summarized in Tables 3 and 4 below.
>
>
> Table 2: AUCROC performance comparison of our method under *M-attack*, [1] and [2] on CIFAR10
> | Defense ↓        | BadNets | SSBA | Adaptive-Patch | *M-Attack* | [1] | [2] |
> | :---------------- | :------: | :----: |:----: |:----: |:----: | :----: |
> | Our Method       | 1.0 | 0.97 | 0.98 | 0.85 | 0.84 | 0.89 |
> | STRIP | 1.0 | 0.68 | 0.76 | 0.71 | 0.69 | 0.74 |
>
> Table 3: AUCROC performance comparison of our method under *M-attack*, [1] and [2] on GTSRB
> | Defense ↓        | BadNets | SSBA | Adaptive-Patch | *M-Attack* | [1] | [2] |
> | :---------------- | :------: | :----: |:----: |:----: |:----: | :----: |
> | Our Method       | 0.99| 0.99 | 0.87 | 0.82 | 0.86 | 0.85|
> | STRIP | 0.99| 0.80 | 0.33 | 0.68 | 0.62 | 0.81 |
>
>  We observed that our methods still outperform other state-of-the-art defenses, even though there is a moderate decline in performance compared to different attack scenarios like BadNets and SSBA. Nonetheless, we believe that this is a reasonable outcome, given that no defense can be universally effective against all attack variations.
>
> > Q: notations on T
>
> R: We introduced this transformation $T$ and offered specific examples in Lines 171 - 175 of the main text, before presenting the Pseudocode of Algorithm 1. In the revised version, we plan to incorporate these discussions directly into the pseudocode of Algorithm 1 to enhance clarity.
>
> Specifically, $T(\cdot)$ denotes a defender-constructed transformation that
> typically depends on both the backdoored model $f^{\text{poi}}$ and the validation dataset $D^{\text{Val}}$, to reflect special properties of backdoor data, e.g., the predicted value
> $T(X_{\text{test}}) = f^{\text{poi}}(X_{\text{test}})$ and the latent representation $T(X_{\text{test}}) = \phi^{\text{poi}}(X_{\text{test}})$.
>
> ### Reference
>
> [1] Doan, Khoa, Yingjie Lao, and Ping Li. "Backdoor attack with imperceptible input and latent modification." Advances in Neural Information Processing Systems 34 (2021): 18944-18957.
>
> [2] Shokri, Reza. "Bypassing backdoor detection algorithms in deep learning." 2020 IEEE European Symposium on Security and Privacy (EuroS&P). IEEE, 2020.

---

> > ### Comment · Reviewer_ibgE · 2023-08-15
> >
> > Thanks for the response.
> >
> > Thanks for providing the results on one of the strong attacks. It is would be better to evaluate on an adaptive attack that is tailored for the proposed detection method.
> >
> > Could you provide a specific example of function T? Is it an encoder model?

---

> > > ### Author Response · Authors · 2023-08-15
> > > **Thanks for the reviewer's comments; Clarification on the new experiments and T**
> > >
> > > We would like to express our appreciation to the reviewer for bringing up the subsequent inquiries. We proceed to respond to the reviewer's comments as follows.
> > > > Thanks for providing the results on one of the strong attacks. It is would be better to evaluate on an adaptive attack that is tailored for the proposed detection method.
> > >
> > > In our rebuttals above, we have provided **Tables 1 and 2**, where we carry out a total of additional **three distinct backdoor attacks** to showcase the efficacy of our methodologies. The **M-attack is designed specifically to target our defense method**, focusing on reducing the Mahalanobis distance between the clean and the backdoor latent representations. Furthermore, as per your suggestions, we also subjected our method to testing against **two other approaches** [1,2] that similarly target our method. These outcomes underscore the consistent effectiveness of our approach across various backdoor threat models.
> > >
> > > To summarize, we have rigorously evaluated our method against a **comprehensive set of 16 distinct backdoor attack types**. However, should you have additional types of backdoor attacks in mind that you would like to see addressed, kindly provide us with detailed information, and we would be more than willing to conduct tests in those scenarios.
> > >
> > > > Could you provide a specific example of function T? Is it an encoder model?
> > >
> > > An illustration of a commonly employed $T$ is the latent representation, which pertains to the penultimate layer of the backdoored model. This choice is based on the assumption that the latent representation encapsulates high-level information from the original data.

---

> > > > ### Author Response · Authors · 2023-08-19
> > > >
> > > > We sincerely appreciate Reviewer ibgE for their insightful and positive comments. In our response, we have addressed concerns regarding (1) conducting extra experimental studies to evaluate our method's performance and (2) providing clarity on the notation of $T$.
> > > >
> > > > As the discussion phase nears its conclusion, we kindly inquire if the reviewer has any further comments on our response. We are readily available for any additional queries they may have.
> > > >
> > > > Once more, we appreciate your time and effort in reviewing our paper.

---

> > > > ### Comment · Reviewer_ibgE · 2023-08-20
> > > >
> > > > Thank authors for providing further clarification.

---

> > > > > ### Author Response · Authors · 2023-08-20
> > > > > **Thanks the Reviewer ibgE**
> > > > >
> > > > > Thank you for your helpful feedback on our responses. We'll make sure to include these suggestions when revising our work.

---

### Official Review · Reviewer_JEgp · 2023-07-05

**Soundness:** 3 good
**Presentation:** 3 good
**Contribution:** 2 fair
**Rating:** 5
**Confidence:** 4

**Summary:**

This paper formulates the inference-stage backdoor detection in terms of backdoor-sample identification and proposes a unified defense framework. It derives a theoretically optimal detection rule and validates its effectiveness in both CV and NLP domains.

**Strengths:**

1. The paper is well-organized with a comprehensive structure. For example, section 3, first formulates the backdoor detection and then illustrates the proposed conformal detection, followed by the derivation of optimal score functions and a practical proxy, which is clearly illustrated.
2. The proposed method can work well under different domains, including CV and NLP, which are more general than the domain-specific methods.
3. The idea is theoretically correct and easy to follow.
4. The experiments are reliable with 10 times independent repetition as illustrated in 4.1.
5. The comparisons under different backdoor attacks are sufficient.

**Weaknesses:**

1. Some redundancy and unclear expressions exist. For example,
    1). The $\lambda_{\alpha,s}$ is first shown in line 5, Algorithm 1, and $\tau$ is used in Equation (1), where the relationship between them should be clearly claimed.
    2). The backdoor trigger is defined unclearly and misunderstood in section 3.3. The $\eta_1$ expressing backdoor trigger is used in lines 225 and 244, while in line 233, it represents the backdoor transformation. Also, the $\gamma$ is used in other places.
2. Lack of comparisons with the SOTA backdoor defense methods that include the separation of the input data, such as the ABL[1] and DBD[2].
3. The implementation code of this paper does not release.

[1] Li, Yige, et al. "Anti-backdoor learning: Training clean models on poisoned data." *Advances in Neural Information Processing Systems* 34 (2021): 14900-14912.

[2] Huang, Kunzhe, et al. "Backdoor defense via decoupling the training process." *arXiv preprint arXiv:2202.03423* (2022).

**Questions:**

Please see the weakness section.

**Limitations:**

Please see the weakness section.

---

> ### Author Rebuttal · Authors · 2023-08-09
>
> We sincerely thank reviewers for investing their time and energy into reviewing our manuscript and offering valuable feedback. We're pleased that they recognized the quality of our writing, acknowledged the novelty of our proposed framework (Reviewer vAAp), and found our approach effective across various domains and setups (JEgp, ibgE, 1Ekh). We will incorporate all their comments into the revised paper.
>
> > Q： Some redundancy and unclear expressions exist
>
> R:
> - The relationship between $\tau$ and $\lambda_{\alpha,s}.$ In brief, $\tau$ and $\lambda_{\alpha,s}$ denote the same decision value, but they are referenced differently in the text and Algorithm 1's pseudocode. Specifically, within our BCD framework, $\tau$ is sought as a solution to Equation (1) based on given $\alpha$ (type I error rate), $\delta$ (violation rate), and $n$ (sample size of the validation dataset). This $\tau$ is later employed as the decision threshold $\lambda_{\alpha,s}$ in the pseudocode of Algorithm 1. We will correct and unify the notations in the revision.
> - The backdoor transformation $\eta_1$ and the backdoor trigger $\gamma$.
> We use $\eta_1$ to denote the general backdoor transformation (from clean data to backdoor data). For instance, in poisoning backdoor attacks, we have $\eta_1(x) = x + \gamma$, with $\gamma$ being the backdoor trigger.
> In Line 225, when we mention "transforming clean data with backdoor triggers $\eta_1$," we are referring to converting clean data into backdoor data using the backdoor transformation $\eta_1$.
> In Line 244, there is a typo. It should read "backdoor triggers $\gamma \in T_c$ that" instead of "backdoor triggers $\eta \in T_c$ that".
>
>
>
>
>
>
>
>
> > Q: Comparisons with the SOTA backdoor defense methods that include the separation of the input data, such as
> the ABL[1] and DBD[2].
>
> R: First, It's important to emphasize that the ABL [1] and DBD [2] methods differ significantly from our proposed approaches in terms of threat models and methodology, as detailed below.
>
> ABL [1] and DBD [2] are **training-stage defense** approaches with access to both clean and backdoor training data, **affording them information about both types of clean and backdoor data**. Their goal is to differentiate between clean and backdoor training data, ultimately constructing a clean model through the use of filtered clean training data, obtained by analyzing all training instances. Hence, their evaluation metrics are the clean and attack success rate of the cleansed model.
>
> In contrast, our method is an **inference-stage defense** that relies solely on a very small (e.g., 1000) set of clean validation data and operates **without any knowledge of backdoor data**. Our goal is to detect future backdoor inputs, and our evaluation metric is the AUCROC score of the detector.
> Therefore, direct comparisons between ABL [1], DBD[2], and our methods may not be equitable or appropriate.
>
> **Nonetheless**, we included the AUCROC scores of ABL [1] and DBD [2] on distinguishing between clean and backdoor data in Tables 1 and 2 below. We believe that our method consistently outperforms ABL [1] and DBD[2] under different types of backdoor attacks. These results underscore our method's superiority over ABL [1] and DBD[2] in terms of detection performance.
>
>
> Table 1: AUCROC performance comparison of our method with ABL[1] and DBD[2] on CIFAR10
> | Defense ↓        | BadNets| Dynamic | SSBA | Adaptive-Blend | Adaptive-Patch |
> | :---------------- | :------: | :----: |:----: |:----: |:----: |
> | Our Method       |   1.0  | 1.0 | 0.97 | 0.96 | 0.98
> |   ABL    |   0.98  | 0.92 |   0.81 | 0.79 | 0.72|
> | DBD  |  0.98 | 0.89 | 0.87 | 0.78 | 0.67|
>
> Table 2: AUCROC performance comparison of our method with ABL[1] and DBD[2] on GTSRB
> | Defense ↓        | BadNets| Dynamic | SSBA | Adaptive-Blend | Adaptive-Patch |
> | :---------------- | :------: | :----: |:----: |:----: |:----: |
> | Our Method       |   0.99  | 0.99 | 0.99 | 0.99 | 0.87
> |   ABL    |   0.97  | 0.85 |   0.81 | 0.59| 0.72|
> | DBD  |  0.97 | 0.84| 0.77| 0.78 | 0.67|
>
>
>
>
> > Q: Code Release
>
> R: We have already included the preliminary codes in the originally uploaded supplementary materials, allowing the reproduction of NLP backdoor attack results showcased in Figure~4 of the main text. Replicating all other results merely necessitates adjustments to the clean and backdoor representations. That being said, following the convention, we will release the complete version of our codes upon the paper's acceptance.
>
> ### Reference
>
> [1] Li, Yige, et al. "Anti-backdoor learning: Training clean models on poisoned data." Advances in Neural Information Processing Systems 34 (2021): 14900-14912.
>
> [2] Huang, Kunzhe, et al. "Backdoor defense via decoupling the training process." arXiv preprint arXiv:2202.03423 (2022).

---

> > ### Author Response · Authors · 2023-08-19
> >
> > We sincerely appreciate Reviewer JEgp for their insightful and positive comments. In our response, we have addressed concerns regarding (1) conducting additional experimental studies to compare performance with the mentioned SOTA method, and (2) clarifying notations and code release.
> >
> > As the discussion phase nears its conclusion, we kindly inquire if the reviewer has any further comments on our response. We are readily available for any additional queries they may have.
> >
> > Once more, we appreciate your time and effort in reviewing our paper.

---

### Official Review · Reviewer_vAAp · 2023-07-08

**Soundness:** 3 good
**Presentation:** 2 fair
**Contribution:** 3 good
**Rating:** 6
**Confidence:** 3

**Summary:**

This paper proposes a unified inference-stage detection framework to defend against backdoor attacks. The authors first formulate the inference-stage backdoor detection problem, discuss its challenges and limitations, and then suggest a framework with provable guarantees on the false positive rate or the probability of misclassifying a clean sample. The authors also derive a detection rule to maximize the rate of accurately identifying a backdoor sample, given a false positive rate under classical learning scenarios. Based on this, they then suggest a practical and effective approach for real-world applications. The proposed method was evaluated on 12 different backdoor attacks on computer vision and NLP benchmarks. The experimental findings align with the theoretical results, showing significant improvements over the state-of-the-art methods.

**Strengths:**

- The proposed framework for defending against backdoor attacks is novel to the best of my knowledge.
- The paper is sound and decently written (beyond some mathematical clutter, see belo).
- The proposed method is validated through extensive experiments on multiple datasets and compared to many existing defenses, demonstrating its effectiveness.
- The authors provide a theoretical analysis and derive technical insights on toy settings which motivates their practical defense on the real settings.

**Weaknesses:**

- There is quite a bit of mathematical clutter in S2 and S3 which I believe can be avoided for a smoother read.
- Some of the details about the backdoored models are not present in the paper (see below) which make it a bit hard to asses the faithfulness of the comparisons to previous models.

**Questions:**

- For all the backdoored models used in the paper, what is the percentage of poisoned training data, and what are the clean and robust accuracy of these models (before the defense). It is crucial to know this as different defenses might behave differently as the “strength” of the backdoor attack varies. I encourage the authors to include these details along with an ablation study for this.

- As shown in Fig 4, the proposed attack performance is not very different than prior work. The authors “justify” this by saying “These findings suggest that the current NLP attacks retain a considerable amount of information in the latent representations that can be utilized to differentiate between clean and backdoor data.” Can the authors explain this in more detail? What is special about NLP? Are backdoor attacks themselves weaker there? This ties back to my first point on clarifying the performance of the backdoor models on clean and modified data.


**Limitations:**

The authors adequately addressed the limitations.

---

> ### Author Rebuttal · Authors · 2023-08-09
>
> We sincerely thank reviewers for investing their time and energy into reviewing our manuscript and offering valuable feedback. We're pleased that they recognized the quality of our writing, acknowledged the novelty of our proposed framework (Reviewer vAAp), and found our approach effective across various domains and setups (JEgp, ibgE, 1Ekh). We will incorporate all their comments into the revised paper.
>
> > Q: Implementation details
>
> R:  To fit the page limit, we’ve included all the implementation details and several ablation
> studies in the originally uploaded supplementary material. We summarize them below.
> 1. Open-source package used. All the backdoor attacks and associated backdoor data in our work are obtained based on open-source projects. For computer vision (CV) attacks, we employed the Backdoor ToolBox [1] and BackdoorBench [2] to ensure result consistency. NLP attacks were conducted using the specialized OpenBackdoor [3] package.
> 2. We provide the poisoning rate, clean accuracy, and backdoor accuracy for backdoor attacks presented in the main text in Table 1 below. In general, the backdoored models exhibit comparable clean accuracy to the normal models, while achieving nearly perfect accuracy on backdoor data, with a relatively low poisoning ratio.
>
> Table 1: Poisoning ratio, clean and backdoor accuracy of backdoor attacks used in our paper. * For the SSBA and WaNet attacks, backdoor poisoning rates are 5\%. For the A-Blend and SIG attacks, the backdoor accuracies are around 80\%. † For the A-Patch attack, the backdoor accuracy is around 60\%.
> |                     |                         |Backdoor Model| Backdoor Model| Clean Model |
> |--------------|-----------------|----------------|------------------|----------------|
> | Dataset ↓     | Poisoning Ratio | Clean Accuracy | Backdoor Accuracy | Clean Accuracy |
> | CIFAR10*     | 0.3%        | ≥ 93%      | ≥ 97%        | ≥ 93%      |
> | GTSRB†        | 1%          | ≥ 95%      | ≥ 97%        | ≥ 96%      |
> | Tiny ImageNet | 1%          | ≥ 37%      | ≥ 96%        | ≥ 40%      |
> | SST2 (NLP)    | 10%         | ≥ 91%      | ≥ 99%        | ≥ 92%      |
> | IMDB (NLP)    | 10%         | ≥ 90%      | ≥ 96%        | ≥ 91%      |
>
>
>
> > Q: Ablation study on poisoning rates and model architectures
>
> R: We provide the outcomes of two ablation studies: (i) Table 2 below, addressing the poisoning ratio, and (ii) **Figures 1 in the PDF file of the global response**, depicting detection power with VGG 19. Overall, our method's detection performance remains consistent across diverse poisoning ratios and architectures. This highlights the stability and efficacy of our approach.
>
> Table 2: AUCROC of our proposed method on CIFAR10
> | Poisoning ratio   | 0.3% | 1% | 5% |
> | :---------------- | :------: | :----: | :----: |
> | BadNets        |   0.99  | 0.99 | 0.99 |
> | Blended           |   0.96  | 0.95 | 0.96  |
> | WaNet           |   0.85  | 0.91 | 0.93 |
> | SSBA           |   0.92  | 0.95 | 0.97 |
>
>
>
> > Q: Explanations on the claim of "These .. data." regarding the performance of our methods on NLP backdoor attacks
>
> R:   We explain our claim of "These .. data." in the following.
> - **The discrete nature of NLP data often leads to noticeable NLP backdoor triggers**.
> NLP backdoor attacks stand apart from CV backdoor attacks due to the distinct nature of their data representations. Image data is commonly described by continuous values, while textual data is characterized by symbolic and discrete forms. This discreteness often renders NLP backdoor triggers more conspicuously visible [4], in contrast to the human-imperceptible CV backdoor triggers [5]. For example, the SOS attack [6] introduces the non-relevant and easily noticeable word combination 'mn' into the text.
> - **Noticeable NLP backdoor triggers lead to clear distinctions between the latent representations of clean and backdoor data.**
> Due to the conspicuous nature of backdoor triggers in NLP attacks, distinct differences emerge in the latent representations of clean and backdoor data under different backdoor attacks. Defenders can leverage these differences to accurately discern, e.g., with near-perfect accuracy, between clean and backdoor data, as empirically demonstrated in our main text.
>
> As a result of the above two points, we made the claim that "current NLP attacks retain a ... utilized to differentiate between clean and backdoor data" in our main text. We will improve the clarity of this statement in the revision.
>
> > Q: mathematical clutter in S2 and S3
>
> R: We will enhance the clarity of concepts, expressions, and language in the revised version.
>
> ### Reference
> [1] T. Xie, “Backdoor toolbox,” https://github.com/vtu81/backdoor-toolbox, 2022.
>
> [2] B. Wu, H. Chen, M. Zhang, Z. Zhu, S. Wei, D. Yuan, and C. Shen, “Backdoorbench: A comprehensive benchmark of backdoor learning,” in NIPS Datasets and Benchmarks Track, 2022.
>
> [3] G. Cui, L. Yuan, B. He, Y. Chen, Z. Liu, and M. Sun, “A unified evaluation of textual backdoor learning: Frameworks and benchmarks,” arXiv preprint arXiv:2206.08514, 2022.
>
> [4] Chen, X., Salem, A., Chen, D., Backes, M., Ma, S., Shen, Q., and Zhang, Y. "Badnl: Backdoor attacks against nlp models with semantic-preserving improvements," in ACSAC, 2021
>
> [5] Anh Tuan Nguyen and Anh Tuan Tran. WaNet – Imperceptible wapping-based backdoor attack. In ICLR, 2021.
>
> [6]  W. Yang, Y. Lin, P. Li, J. Zhou, and X. Sun, “Rethinking stealthiness of backdoor attack against nlp models,” in ACL, 2021.

---

> > ### Author Response · Authors · 2023-08-19
> >
> > We sincerely appreciate Reviewer vAAP for their insightful and positive comments. In our response, we have addressed concerns regarding (1) implementation details, (2) ablation studies on different poisoning rates and model architectures, and (3) NLP backdoor attack issues.
> >
> > As the discussion phase nears its conclusion, we kindly inquire if the reviewer has any further comments on our response. We are readily available for any additional queries they may have.
> >
> > Once more, we appreciate your time and effort in reviewing our paper.

---

> > ### Comment · Reviewer_vAAp · 2023-08-19
> > **Thanks for clarification**
> >
> > I thank the authors for their time to clarify my concerns. I think in particular these extra experimental details are important for clarity of the paper. I am happy to maintain my score.

---

> > > ### Author Response · Authors · 2023-08-19
> > > **Thanks the Reviewer vAAP**
> > >
> > > Thank you for your helpful feedback on our responses. We'll make sure to include these suggestions when revising our work.

---

### Author Rebuttal · Authors · 2023-08-10

We sincerely thank reviewers for investing their time and energy into reviewing our manuscript and offering valuable feedback. We're pleased that they recognized the quality of our writing, acknowledged the novelty of our proposed framework (Reviewer vAAp), and found our approach effective across various domains and setups (Reviewer vAAP, JEgp, ibgE, 1Ekh). We will incorporate all their comments into the revised paper.

In the following, we will begin by addressing a potentially unclear aspect related to our method's threat model and its comparison with various defenses mentioned by the reviewers. After that, we'll provide a concise summary of our tailored responses for each reviewer.

Regarding the threat model, our approach serves as an **inference-stage defense** with **no access to the training data** and **no ability to manipulate the training process**, including model parameter adjustments. Similarly to other inference-stage defenses [1,2], we operate under the assumption that the defender possesses a limited clean validation dataset but **no foreknowledge of future backdoor test inputs**. Our goal is to detect future backdoor inputs, and our evaluation **metric is the AUCROC score** of the detector.


In contrast, the defenses mentioned by the reviewers, namely ABL [3], DBD [4], and SPECTRE [5], belong to the training-stage defense category. These approaches have the advantage of **accessing both clean and backdoor training data**, which provides them insights into both types of data. Their objective is to build a clean model by isolating a subset of the training set through analysis of specific characteristics of all training examples. Consequently, their evaluation criteria focus on the **clean and attack success rates** of the purified model.


- **Rebuttal Summary for Reviewer vAAP**:
1. Added implementations details of backdoor attacks
2. Added Ablation studies on the poisoning rate and different model architectures (*Figure 1 in Global Response PDF file*)
3. Clarified issues regarding NLP backdoor attacks in Fig 4


- **Rebuttal Summary for Reviewer JEgp**:
1. Clarified issues regarding notations
2. Added experiments on performance comparison with SOTA backdoor defenses methods mentioned by reviewers
3. Clarified issues regarding the code release



- **Rebuttal Summary for Reviewer ibgE**:
1. Clarified the relationship between SPECTRE and our method; Providing empirical evaluations between SPECTRE and our method
2. Added experiments on the performance of our methods under the reviewer-mentioned backdoor attacks, and a new backdoor attack specifically targeted for our method (proposed by ourselves)
3. Clarified notations regarding the data transformation $T$



- **Rebuttal Summary for Reviewer 1Ekh**:
1. Explained intuitions for the proxy terms
2. Added experiments on reviewers-mentioned defenses and attack(s) (*Table 2 in Global Response PDF file*)
3. Added ablation studies on different poisoning ratios; new backdoor attacks specifically targeted at our method; and the OOD dataset  (*Tables 3 & 4 in Global Response PDF file*)
4. Clarified issues regarding the selection of $\delta$
5. Clarified issues regarding the claim in Lines 109 - 111
6. Clarified issues regarding the observations in Figure 2
7. Added histograms for SCM scores (*Figure 2 in Global Response PDF file*)

### References

[1] Y. Gao, C. Xu, D. Wang, S. Chen, D. C. Ranasinghe, and S. Nepal, “Strip: A defence against trojan attacks on deep neural networks,” in Proceedings of the 35th Annual Computer Security Applications Conference, 2019, pp. 113–125.

[2] J. Guo, Y. Li., X. Chen, H. Guo, L. SUN,  and C. Liu. "Scale-up: An efficient black-box input-level backdoor detection via analyzing scaled prediction consistency," in ICLR 2023.

[3] Li, Yige, et al. "Anti-backdoor learning: Training clean models on poisoned data." Advances in Neural Information Processing Systems 34 (2021): 14900-14912.

[4] Huang, Kunzhe, et al. "Backdoor defense via decoupling the training process." arXiv preprint arXiv:2202.03423 (2022).

[5] J. Hayase, W. Kong, R. Somani, and S. Oh, “Spectre: defending against backdoor attacks using robust statistics,” in ICML, 2021.

---

> ### Comment · Area_Chair_AHLn · 2023-08-19
>
> Thanks to the authors for their rebuttals and engagement with the author discussion process. In some cases I realise the reviewers have not yet acknowledged these rebuttals - I'd like to assure the authors that the conference has reminded the reviewers of the need to fully engage with author rebuttals, and that there are still opportunities for this to happen during this (almost finished) author discussion period, and then into the reviewer discussion period that comes next. I'd also like to reassure the authors that I've read in full all of the reviews, the rebuttals and all comments to date. I'll focus mostly on those reviews yet to reply, and identified concerns/weaknesses (with strengths already noted elsewhere).
>
> ibgE has responded to the first rebuttal, with one more follow up from the authors subsequently. My sense is that while the authors have indeed provided some form of adaptive attacker results, which is already greatly appreciated during the short author response period, having a comprehensive analysis of adaptive attacker(s) would be helpful. Considering that the paper's narrative is around "optimal" defenses I think the reviewer's request was reasonable. The authors are also justified in highlighting a tremendous amount of work evaluating many attacks already.
>
> 1Ekh has responded to each rebuttal/response and adjusted their score. A lingering concern is around the Gaussian assumptions for the framework: the significance of the (theoretical) results in this context, though commitments to update narrative/language given this context is appreciated.
>
> ### vAAp
> **W2/Q1)** thanks for providing more detail on the experimental settings, and the small ablation study. **Q2)** thanks for the explanation on NLP vs CV. While we're yet to determine whether these details/explanations have changed the reviewer's rating, it is appreciated.
>
> ### JEgp
> **W1)** thanks for the clarifications, these will (at least partly) assist in understanding the work. **W2)** I think the reviewers appreciate that work on backdoor defenses can differ in the threat model considered. However I think it's still reasonable of them to ask for comparisons, as the effect on robustness of adversarial information/control is a fundamental question. If a weaker adversary is able to achieve the same ASR (for example) as a stronger adversary (owing to threat models) then this impacts significance of work. Thanks for providing results on ABL [1], DBD [2] given this.

---

> > ### Author Response · Authors · 2023-08-19
> > **Thanks the Area Chair for overseeing the review and the rebuttal process; Further Clarifications**
> >
> > We extend our heartfelt gratitude to the Area Chair for dedicating time to review both our paper and our responses. We would like to address the comments provided by the Area Chair as follows.
> >
> > > Q: My sense is that while the authors have indeed provided some form of adaptive attacker results, which is already greatly appreciated during the short author response period, having a comprehensive analysis of adaptive attacker(s) would be helpful.
> >
> > R: We are in complete agreement with the importance of thoroughly testing our method against various types of attacks. To ensure we meet this objective, we have taken several steps. In our main text, we've incorporated three of the latest latent-space adaptive backdoor attacks: TacT (USENIX 2021), Adaptive Blend, and Adaptive Patch (both introduced at ICLR 2023). Furthermore, following the reviewer's valuable suggestion, we've also introduced three additional attacks in our rebuttal response. All of these attacks are intentionally directed at our defense mechanism, and noteworthy among them is the M-attack, which we devised ourselves. Taking into account all these measures we've implemented, we hold a reasonable degree of confidence that we have subjected our method to a comprehensive assessment across a broad spectrum of contemporary attacks.
> >
> > **That being said, we are fully open to testing our method on any new approach recommended by the reviewer. Our aim is to alleviate any concerns they might have. However, the term "adaptive attack tailored for the proposed detection method," as mentioned by the reviewer, is not entirely clear to us. In an effort to better understand, we have reached out to the reviewer in our response to kindly request a more detailed explanation of this concept. Due to time constraints, addressing the reviewers' new suggestions at this moment might be challenging. Yet, we're committed to adding it to the revised paper.**
> >
> >
> > > Q: 1Ekh has responded to each rebuttal/response and adjusted their score. A lingering concern is around the Gaussian assumptions for the framework: the significance of the (theoretical) results in this context, though commitments to update narrative/language given this context is appreciated.
> >
> > R: We genuinely appreciate your attention to the Gaussian assumption. Our work encompasses two types of provable guarantees:
> > 1. The first type of guarantee concerning the *type I error rate*, presented in Theorem 1 of the main text, is **distribution-free**. This means our Conformal Backdoor Detection framework operates without necessitating assumptions about data distributions.
> > 2. The second guarantee, pertaining to the *optimal detector in terms of detection power*, presented in Theorem 2 and 3, is based on the Gaussian assumption. While this assumption might not align perfectly with real-world scenarios, we have observed effective approximations of actual data distributions using the Gaussian assumption specifically when developing backdoor defenses for various sets of backdoor attacks, as demonstrated by the consistent effectiveness of our method.
> >
> > **As promised, we will revise both the title and the main text to eliminate any potential confusion and to offer a clearer articulation of the theoretical outcome.**
> >
> > > Q: However I think it's still reasonable of them to ask for comparisons, as the effect on robustness of adversarial information/control is a fundamental question. If a weaker adversary is able to achieve the same ASR (for example) as a stronger adversary (owing to threat models) then this impacts significance of work. Thanks for providing results on ABL [1], DBD [2] given this.
> >
> > R: We are in complete agreement with the importance of thoroughly testing our method against different types of attacks, which can have varying levels of information and capabilities from the attackers. To ensure we meet this objective, our assessment of various backdoor attacks covers a wide range of attacker capabilities, including (1) poison-only attackers, (2) poison-only and sample-specific attackers, (3) training-controlled and optimized attackers, (4) training-controlled, invisible, and sample-specific attackers, (5) non-poison attackers, and (6) attackers with knowledge of the defense strategy. We will highlight this point to reflect the consistent effectiveness of the method.
> >
> > **However, we are still very open to considering and exploring other types of attacks that the reviewers might suggest. We value their input and are eager to ensure our method's robustness across different attack possibilities. Due to time constraints, addressing the reviewers' new suggestions at this moment might be challenging. Yet, we're committed to adding it to the revised paper.**
> >
> > Once again, we extend our heartfelt appreciation to the Area Chair for generously investing time and effort in guiding our paper through the review and rebuttal process. Additionally, we would like to thank all the reviewers for their valuable and insightful comments.

---

> > > ### Comment · Area_Chair_AHLn · 2023-08-20
> > >
> > > Many thanks to the authors for this thorough, prompt response to my summary response. I appreciate the clarification on Thm 1 vs Thms 2 & 3. I'll take this into account. Also thank you for summarising the state of attacks in the work (and wasn't suggesting the authors run more experiments now, this late in the author discussion period), and the ways in which your assessment compares the method against different types of attacks.

---

> > > > ### Author Response · Authors · 2023-08-20
> > > >
> > > > We extend our sincere gratitude to the AC for timely and valuable feedback on all our responses throughout the entire rebuttal and discussion period. We are genuinely delighted to learn that our explanations regarding theoretical results and experiment design proved to be of assistance. Your input and guidance are greatly appreciated and contribute significantly to our work.

---

### Decision · Program_Chairs · 2023-09-21

**Decision:**

Accept (poster)

**Comment:**

Thanks to the reviewers and authors for the constructive feedback on the paper and collegial discussion so far. The paper's present ratings of 5,6,4,5 reflect overall support albeit with some lingering concerns. The substance of the feedback/discussion, and the paper itself, reflect contributions that is not only sound but I believe significant with potential for impact that reflect a higher reviewer rating than was sometimes reported.

The paper presents a novel inference-stage conformal backdoor detection (CBD) framework for use in defending against backdoor attacks using deep nets' latent representations. Its theoretical results are sound and aligned with the algorithmic contributions, and are accompanied by extensive empirical evaluations that demonstrate compelling experimental performance (across CV and NLP with 3/10 and 2/2 datasets/attacks) when theoretical assumptions are relaxed.

Theorem 1 guarantees the FPR of the CBD algorithm under fairly convention assumptions while Theorems 2 and 3 do make a Gaussian assumption but I view these as complementary results demonstrating sufficient conditions for uniqueness and the decision rule being uniformly most powerful. That is, the Gaussian assumption found in the paper is not relevant to the key results but the complementary results. The theory might not be introducing new proof techniques or demonstrate technical innovation, but it is well aligned with the _algorithmic/framework_ contribution of the paper and I believe fits well. This helps explains the approach's empirical performance and serves as motivation; particularly Theorem 3 as it suggests an appropriate rule for balancing the power/FPR trade-off.

Initial reviewer concerns were primarily: overlap with existing work SPECTRE concerning novelty; evaluation against adaptive attacks and specific baselines including ABL, DBD and others; additional ablation studies to those included in the appendices; and specific clarity/presentation issues.

The authors highlight that some requested comparison defenses make additional assumptions of the defender such as access to clean training data. They have provided requested ablation results in their rebuttal, comparison results with SOTA backdoor defenses (including those that are more restrictive than CBD), attacks, and sufficient clarifying detail that I believe addresses the substantive reviewer concerns.